# Hydrogen embrittlement in metallic nanowires

Sheng Yin[1,4], Guangming Cheng [2,4], Tzu-Hsuan Chang [2], Gunther Richter[3], Yong Zhu[2,5] & Huajian Gao [1,5]

Although hydrogen embrittlement has been observed and extensively studied in a wide variety of metals and alloys, there still exist controversies over the underlying mechanisms and a fundamental understanding of hydrogen embrittlement in nanostructures is almost non-existent. Here we use metallic nanowires (NWs) as a platform to study hydrogen embrittlement in nanostructures where deformation and failure are dominated by dislocation nucleation. Based on quantitative in-situ transmission electron microscopy nanomechanical testing and molecular dynamics simulations, we report enhanced yield strength and a transition in failure mechanism from distributed plasticity to localized necking in penta-twinned Ag NWs due to the presence of surface-adsorbed hydrogen. In-situ stress relaxation experiments and simulations reveal that the observed embrittlement in metallic nanowires is governed by the hydrogen-induced suppression of dislocation nucleation at the free surface of NWs.

[1] School of Engineering, Brown University, Providence, Rhode Island 02912, USA. [2] Department of Mechanical and Aerospace Engineering, North Carolina State University, Raleigh, North Carolina 27695, USA. [3] Max Planck Institute for Intelligent Systems, Heisenbergstrasse 3, D-70589 Stuttgart, Germany. [4] These authors contributed equally: Sheng Yin and Guangming Cheng. [5] These authors jointly supervised this work: Yong Zhu and Huajian Gao. Correspondence and requests for materials should be addressed to Y.Z. (email: yong_zhu@ncsu.edu) or to H.G. (email: huajian_gao@brown.edu)

The phenomenon that hydrogen atoms in the environment impair the mechanical properties of a range of metals and alloys is a century-old problem generally known as hydrogen embrittlement (HE)[1,2]. A number of HE mechanisms have been proposed in the literature[3,4], including H-enhanced decohesion[5,6], H-enhanced local plasticity (HELP)[7,8], blister/bubble formation[9], interface failure[10,11], and hydrogen-induced superabundant vacancies[12,13]. On the other hand, it has been difficult to validate each of the proposed mechanisms individually, as they are not mutually exclusive and multiple mechanisms could coexist. Continuum theories and simulations have shown that in the presence of hydrogen, the solute drag force exerted on a moving dislocation adds resistance to dislocation motion[14,15] and no shielding of dislocation–dislocation interactions by hydrogen can be found in atomic simulations[16]. Hydrogen segregation to grain boundaries and interactions with vacancies also play important roles in HE[17–19]. A recent study suggested that vacancies can interact with hydrogen and lock dislocations motion in aluminum[20]. A chemomechanical origin of hydrogen trapping and segregation at grain boundaries was revealed in recent atomic level studies of HE[21,22].

Most of these existing studies focused on the movement, multiplication, and interaction of dislocations with solute hydrogen in a bulk material and very little is known about whether HE exists also in nanostructures where dislocation nucleation plays a dominant role in deformation mechanisms[23–26]. In bulk materials, softening in pop-in stress during nanoindentation in the presence of hydrogen[27] has not yet been clearly understood. Atomistic simulations revealed that hydrogen-induced swelling of the lattice[28] can contribute to the softening but only to a limited extent. Of particular note is that in atomic simulations, the hydrogen effect on dislocation nucleation can be affected by many factors such as strain rate, nucleation source, and initial defects. Preexisting dislocations and other types of defects cannot be neglected either in such simulations[29,30]. In contrast to the homogeneous nucleation of dislocations under nanoindentation[31], it was found that dislocation emission from a crack tip could be suppressed by the accumulation of hydrogen[32–34].

In spite of the numerous studies of HE in bulk materials, there has been little study on the effect of hydrogen on the mechanical properties of nanoscale materials (e.g., nanowires (NWs)) with a large surface/volume ratio[35–37]. On the other hand, due to complex microstructures often involved in the bulk materials, a fundamental understanding of the HE mechanisms still remains in debate. Studies on simpler material systems such as NWs with well-defined microstructures could provide a new approach and definitive new knowledge to understanding HE. Here, based on a recently developed testing platform combining state-of-the-art microelectromechanical system (MEMS) technology, in-situ transmission electron microscope (TEM) tensile tests and molecular dynamics (MD) simulations, we use metallic NWs as a new platform to investigate HE at the nanoscale. Specifically, we report enhanced yield strength and a transition in failure mode from distributed plasticity to localized necking in penta-twinned Ag NWs in the presence of surface adsorbed hydrogen. In-situ stress relaxation experiments and simulations reveal that the underlying mechanism of HE in NWs is the hydrogen-induced suppression of surface nucleation of dislocations, leading to higher yield strength and failure by localized necking in penta-twinned Ag NWs in a hydrogen environment.

## Results

### NW morphology and in-situ TEM tensile testing.
Penta-twinned Ag NWs were synthesized by a modified polyol process[38]. Based on TEM characterization, the synthesized NWs are straight and uniform in diameter, with a growth direction of < 110 >, as shown in Fig. 1a. The inset in Fig. 1a shows a cross-sectional TEM image of a penta-twinned Ag NW, with five twin boundaries running parallel to the NW length direction. In this study, Ag NWs were immersed in $H_2/Ar$ (molar ratio, 1:1) atmosphere for hydrogen charging (see details in Supplementary Note 1). The relative concentration of hydrogen in the samples was characterized by time-of-flight secondary ion mass spectrometry (SIMS). The increment of hydrogen concentration ($c_H$) in the samples (compared with the as-received, hydrogen-free condition) ranged from 14 to 75 wt p.p.m. (equivalent to surface coverage $\theta_H$ from 0.17 to 0.67 for NW with diameter of 70 nm) as a result of different charging times (12–48 h), as shown in the inset in Fig. 2b (see more details in Supplementary Fig. 1 and Supplementary Table 1). The high concentration of hydrogen measured from the SIMS experiments can be attributed to the surface adsorption of hydrogen on Ag NWs (large surface-to-volume ratio of NWs facilitates the surface adsorption). Hydrogen adsorption on metal surface involves two steps: dissociation of $H_2$ molecule and transport of the chemisorbed hydrogen[39]. Flat and clean Ag surface is known to have a very high energy barrier for hydrogen dissociation and formation of an Ag-H adsorptive bond is endothermic[39–41]. Previous studies revealed that impurity atoms, such as surface and subsurface oxygen species, can promote dissociation of hydrogen molecules[42,43]. In the Ag NWs, the surface is not flat (e.g., roughness and surface steps/defects[44]). These conditions make hydrogen adsorption possible but the activation energy is still very high when compared with other face-centered cubic (FCC) metals such as Cu or Pt[40,41]. The characteristic timescale of adsorption can be hours for this high activation energy of dissociation[43], consistent with our experiments. The relation of surface coverage $\theta_H$ with hydrogen concentration ($c_H$ in wt p.p.m.) is shown in Supplementary Table 1 for the NWs with average diameter of 70 nm. The high concentration of the surface-adsorbed hydrogen atoms and long charging time make it possible for hydrogen atoms to diffuse to the potential nucleation sites (such as surface steps, notches, or vacancies) and be trapped at those sites (see more detailed discussion in Supplementary Notes 1-2 and Supplementary Figs. 2, 3).

We performed in-situ TEM tensile testing of individual NWs using a MEMS-based tensile testing stage (see Supplementary Note 3) that allows accurate measurements of both load and displacement simultaneously, as well as real-time imaging of microstructure evolution during deformation[45,46]. Figure 1b shows the stress–strain responses of Ag NWs with different H concentrations. Figures 1c, d show snapshots of microstructure evolution for typical tensile tests of a hydrogen-free Ag NW and one with $c_H = 75$ wt p.p.m. ($\theta_H = 0.67$). It has been reported that the mechanical properties of penta-twinned Ag NWs are strongly size dependent[24,47]. To focus our attention on the effect of hydrogen, all of the tested NWs in this work were selected to have nearly identical diameter of 71 ± 3 nm (see Methods). As shown in Fig. 1b, with the increase of hydrogen concentration ($c_H$ from 0 to 75 wt p.p.m.), the yielding strength and ultimate strength of the tested NWs increased from 0.95 and 1.02 GPa to 1.54 and 1.59 GPa, respectively, whereas the failure strain (measured between the two deposited local markers) decreased dramatically (from 4.92 to 2.32%), indicating hydrogen-induced embrittlement in the tested NWs. Figure 1c shows that for a hydrogen-free NW, when the applied stress exceeded the yield strength, multiple dislocations were nucleated from the free surface and propagated toward the interior of the NW (Fig. 1c-ii). As the load increased, multiple neckings were observed, leading to large plasticity (Fig. 1c-iii, iv). From the TEM characterization, the maximum reduction of the cross-sectional area was ~15%. In the NW with $c_H = 75$ wt p.p.m.

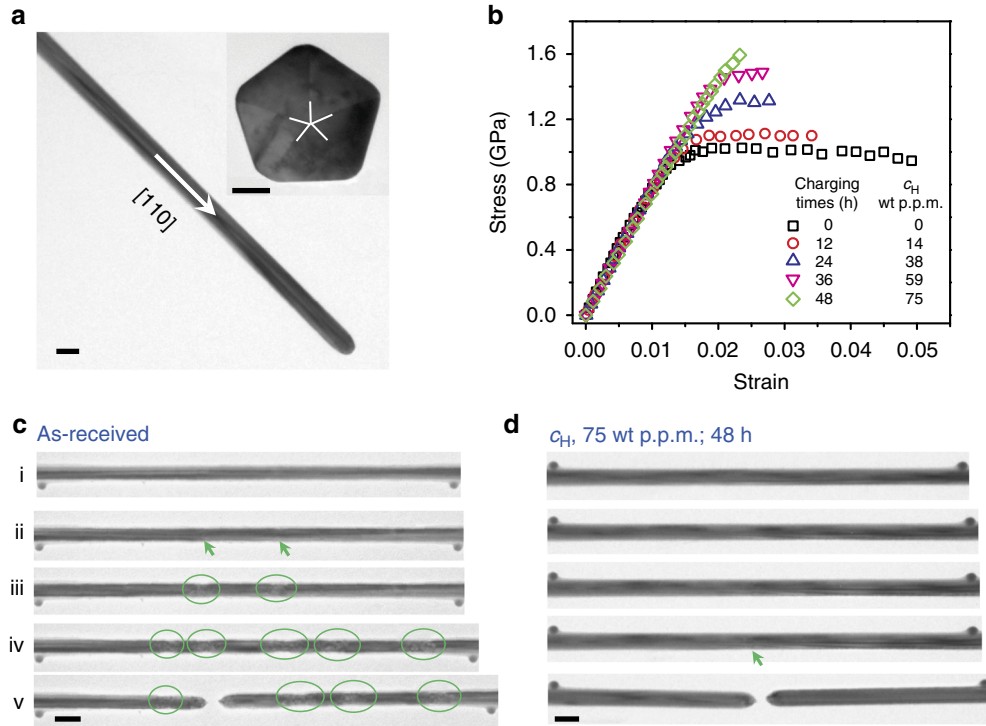

**Fig. 1** Characterization of penta-twinned Ag NWs and TEM tensile tests. **a** TEM image of a penta-twinned Ag NW, which is straight and uniform in diameter with a axial direction of < 110 >. Scale bar in the inset, 20 nm. **b** Stress–strain responses for Ag NWs at different H concentrations. **c** Distributed plasticity in a Ag NW in the absence of hydrogen. The nucleation of dislocations and multiple necks were marked by green arrows and ovals, respectively; i–iv correspond to strains of 0, 1.3, 1.4, and 4.9%. **d** Localized failure of a Ag NW in the presence of hydrogen; i–iv correspond to strains of 0, 1.5, 2.0, and 2.3%. The nucleation of dislocations was marked by green arrow. Scale bars in **a**, **c**, and **d**, 100 nm. The strain rate for in-situ TEM tensile testing was ~0.005%/s

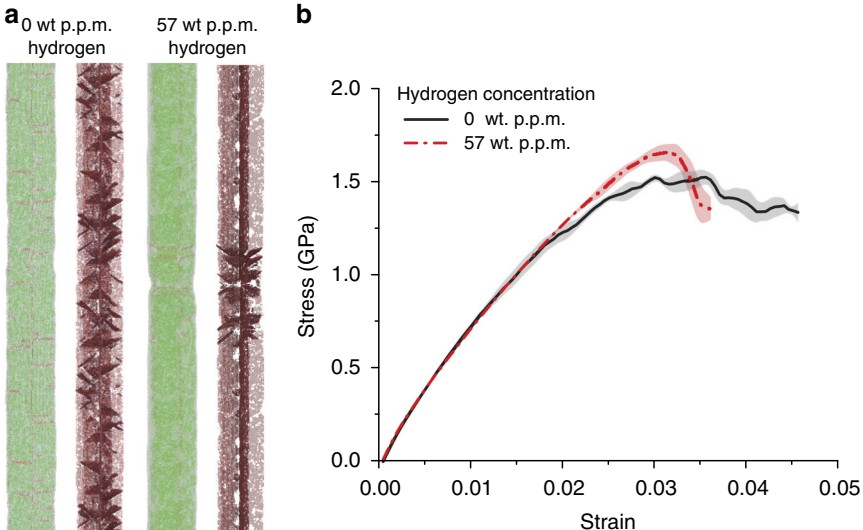

**Fig. 2** MD simulation of tensile testing of penta-twinned Ag NWs. **a** Simulated transition from distributed plasticity in the absence of hydrogen to localized plasticity in the presence of hydrogen. Scale bar, 7 nm. **b** Averaged stress–strain curves of Ag NWs at different hydrogen concentrations with SE bars. Black line corresponds to 0 wt p.p.m. of hydrogen and red dashed line correspond to 57 wt p.p.m. of hydrogen

($\theta_H = 0.67$), there were rare dislocation activities until the applied stress was close to the ultimate tensile strength (Fig. 1d-ii-iv) and the NW fractured before substantial plastic deformation was activated. Multiple dislocation nucleation and propagation were also observed in the NW with $c_H = 38$ wt p.p.m. ($\theta_H = 0.33$, see details in Supplementary Fig. 5). These results suggest that the presence of hydrogen adsorbed on the NW surface and diffusing into potential surface nucleation sources suppressed the nucleation of surface dislocations in the metallic NWs. The increased yield strength has also been found in single crystalline Ag NWs with hydrogen using the same experimental setup (see Supplementary Fig. 6), which can further confirm the suppression effect of hydrogen on surface dislocation nucleation.

**MD simulations of tensile testing**. To reveal the underlying mechanisms behind the observed enhanced yield strength and localized necking behavior of the penta-twinned Ag NWs, we performed a series of MD simulations, the details of which are supplied in the Methods. Dislocation nucleation is very sensitive to pre-existing defects, surface roughness, and other types of nucleation sites[44,48,49]. Previous studies[50] have shown that vacancy defects are present in penta-twinned Ag NWs and can substantially decrease the activation energy for surface dislocation nucleation. In our current model, we hence introduced uniformly distributed vacancies. Hydrogen atoms were initially inserted on the free surface of NW samples at random. Strain rate is known to be of critical importance in MD simulations. To capture the hydrogen effect in simulations, the loading timescale should be comparable to the diffusion timescale of hydrogen, otherwise hydrogen would behave like sessile inclusions and promote dislocation nucleation as a result of lattice distortion. To overcome the timescale issue, a strain rate of $10^6 \text{ s}^{-1}$ and an elevated temperature of 800 K were adopted in our MD simulations. Hydrogen atoms can diffuse into the NWs during the tensile simulations. Detailed discussions of the strain rate and temperature effects are provided in Supplementary Figs. 7–8. Figure 2a shows snapshots of the observed failure modes. In hydrogen-free NWs, dislocation nucleation is evenly distributed along the whole NW, which is consistent with the distributed plastic deformation observed in our experiments. In the presence of hydrogen, localized dislocation nucleation and necking are captured. The stress–strain curves in Fig. 2b show that in the presence of hydrogen, yield strength is higher than that of the hydrogen-free NW, consistent with our experiments. The simulations indicate that hydrogen atoms initially adsorbed on the surface or charged in subsurface regions (Supplementary Fig. 9) can suppress surface dislocation nucleation effectively and lead to HE in the charged NWs. Our simulations show that low strain rates and pre-existing defects are necessary for atomistic modeling of hydrogen effects in nucleation-controlled plasticity. Surface dislocation nucleation is the dominant deformation mechanism in NWs due to dislocation starvation associated with the small size[23–26,51,52]. The distributed plasticity in pristine penta-twinned NWs is a typical example of dislocation nucleation-dominated deformation. The transition from distributed plasticity to localized necking in the presence of hydrogen suggests that, in contrast to previous HE mechanisms in bulk materials and homogeneous dislocation nucleation under nanoindentation, the presence of hydrogen around the surface nucleation sites in NWs increases the activation energy and suppresses surface dislocation nucleation. Specifically, we found that the drop in local activation energy associated with sudden release of a hydrogen-locked site and the interaction between non-uniform hydrogen distribution and surface nucleation sites promote localized plasticity; see Discussion section for details. To validate this hypothesis, we conducted in-situ TEM stress relaxation experiments and further atomistic simulations.

**In-situ TEM stress relaxation testing**. Our previous study reported a dislocation-based stress relaxation behavior in penta-twinned Ag NWs[50], where the stress relaxation is governed by the surface nucleation of dislocations. To illustrate the effect of hydrogen on surface dislocation nucleation, we performed in-situ stress relaxation experiments on NWs charged by hydrogen. During the experiments, the displacement of the MEMS actuator was held constant, whereas the specimen was relaxed as a function of time. Due to the compliance of the loading stage, the load on the specimen decreased while the specimen elongation increased at the same time. Thus, recoverable plasticity[50] was observed in the penta-twinned NWs in the presence of hydrogen and the change in the magnitude of stress relaxation at different hydrogen concentrations reveals an essential role of hydrogen on surface nucleation.

Figure 3a shows stress–strain responses during relaxation tests of three penta-twinned Ag NWs with different concentrations of H inside (charging time 0, 24, and 48 h) in four steps: loading, relaxation, unloading, and recovery. During the loading step, all three NWs exhibited nearly linear responses. During the relaxation step, the stress decreased with time, whereas the strain increased, which is governed by the surface dislocation nucleation; the relaxation magnitude decreased with increasing hydrogen concentration. Figure 3b shows the strain and stress as functions of time for the three NWs at stress levels close to the yield strength. For the hydrogen-free NW, when the initial stress was 0.91 GPa, the NW strain increased monotonically with time from 1.20 to 1.46%, and the stress decreased monotonically from 0.91 to 0.79 GPa. For the 24 h hydrogen-charged sample, the stress decreased from 0.94 to 0.87 GPa and the strain increased from 1.23 to 1.34%. For the 48 h hydrogen-charged sample, the stress decreased from 1.11 to 1.09 GPa and the strain increased from 1.41 to 1.45%, with almost negligible relaxation. The stress relaxation became attenuated as the hydrogen concentration increased. As the stress relaxation is due to sustained surface nucleation of dislocations, the attenuated stress relaxation suggests that the presence of hydrogen has quenched surface nucleation sources in the Ag NWs.

**MD simulations of dislocation nucleation and activation energy**. To further validate the hypothesis that hydrogen can suppress surface dislocation nucleation, atomistic simulations similar to the in-situ stress relaxation experiments were carried out, with vacancies introduced as the initial defects. Different concentrations of hydrogen were initially inserted on the free surface of the NW samples, ranging from 0 to 0.5% (0–46 wt p.p. m.). To simulate the stress relaxation experiments, we first stretched the NW samples to a constant stress and then monitored stress relaxation, while the applied strain was fixed. The simulation results show a significant stress relaxation in a hydrogen-free NW. The discrete stress drops in the stress relaxation profile correspond to (partial) dislocation nucleation events. The right insets in Fig. 3c show stacking faults in the NWs that result from dislocation nucleation events in the twin-boundary separated nanograins. With more hydrogen introduced into the penta-twinned NWs, there is clearly a decrease and delay in stress drops during the relaxation, and fewer dislocations are observed, as illustrated in Fig. 3c. As the hydrogen concentration further increases to ~0.5% (46 wt p.p.m.), no stress relaxation and dislocation nucleation are observed within the simulation timescale. The simulation results clearly show that the presence of hydrogen will delay and suppress dislocation nucleation in penta-twinned Ag NWs. Similar stress relaxation simulations were also performed on a penta-twinned Ag NW with a surface notch instead of vacancies (see Supplementary Fig. 10). Hydrogen was found to accumulate around the surface notch and suppress dislocation nucleation from the notch tip, similar to the previously reported crack tip simulations[33].

In the above stress relaxation simulations, pre-existing defects were introduced as uniformly distributed vacancies in the NWs or as a single surface notch. It is shown that hydrogen can effectively diffuse to the vacancy sites or accumulate around the notch tip and significantly suppress dislocation nucleation. In addition to these two types of defects, surface steps and roughness were also shown to be the preferred sites for dislocation nucleation in NWs[44,48–50]. Minor surface steps/defects can be observed

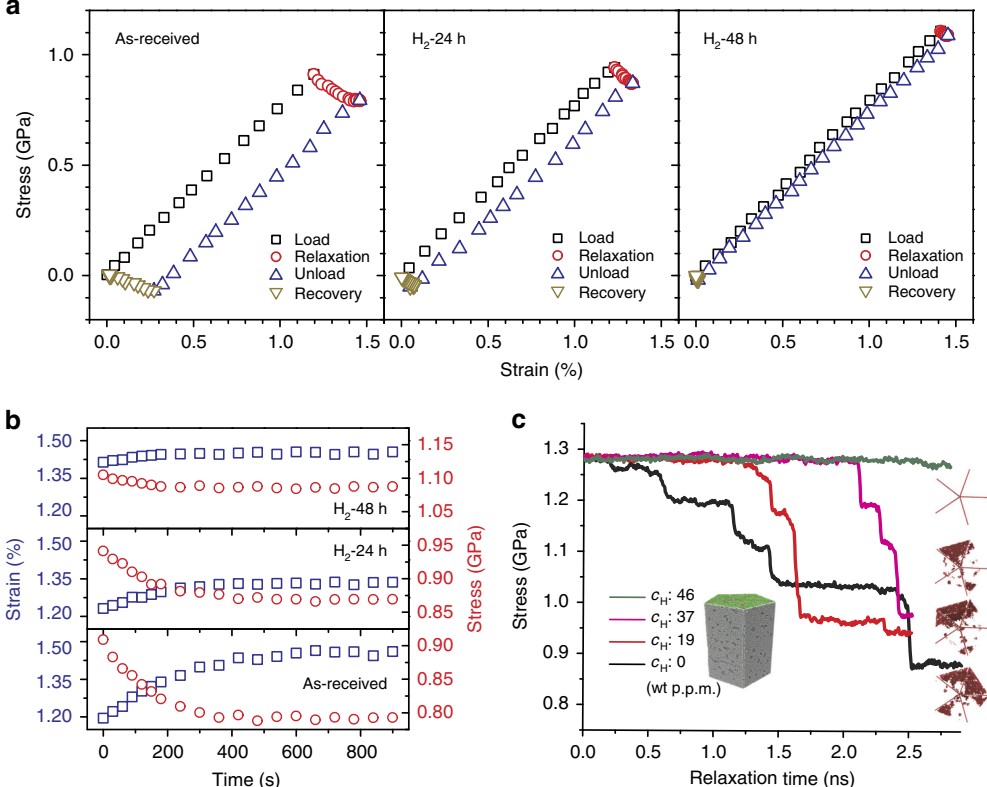

**Fig. 3** In-situ TEM tests and MD simulations of stress relaxation. **a** Stress–strain curves from stress relaxation tests for penta-twinned Ag NWs at different concentrations of hydrogen. **b** Experimentally measured stress relaxation and associated strain evolution curves for penta-twinned Ag NWs at different hydrogen concentrations. **c** MD simulations of stress relaxation in penta-twinned Ag NWs at different hydrogen concentrations $c_H$. Red atoms in the grain show stacking faults in the right insets

through high-resolution TEM on the surface of single crystalline Ag NWs[44]. With a simplified atomic model, we will show shortly that hydrogen can increase the activation energy for surface dislocation nucleation and suppress dislocation nucleation.

In our experiments, the NWs have the axial direction along $<110>$ and free surfaces of (001). To model dislocation nucleation from a surface with roughness or steps/defects[44], we constructed a simplified two-dimensional model, as shown in the inset in Fig. 4e. The tensile direction $<110>$ and out-of-plane direction $<\bar{1}10>$ are periodic. Surface roughness/steps, represented by several surface ledges, were constructed on the (001) free surface; see the Methods section for details. Three different types of surface ledges are considered as potential nucleation sites for surface dislocation. To overcome limitations associated with the diffusion timescale in the simulation, hydrogen atoms were directly inserted into the interstitial sites right next to the surface ledge in sequence. The green stars in Fig. 4a–d represent interstitial positions for the inserted hydrogen atoms and dislocation nucleation sites in the succeeding deformation are also shown. Figure 4e shows the stress–strain curves corresponding to different hydrogen positions and dislocation nucleation sites in Fig. 4b–d. Dislocation nucleation occurs at the inner ledges (easiest for nucleation) at a strain of 4.68% in the absence of hydrogen (Fig. 4a). When the easiest ledges for surface nucleation are charged with hydrogen, these ledges become locked, critical strain for nucleation increases to 5.29%, and nucleation occurs at the neighboring ledges (Fig. 4b). Continued charging of hydrogen causes more surface ledges to be locked (Fig. 4c) and, finally, when all the ledges are charged with hydrogen, the critical strain is increased to 6.54% and dislocation nucleation occurs at a site on the flat surface close to the inner

surface ledge (Fig. 4d). Atomic stress distribution shows that hydrogen atoms close to the surface ledge can substantially relieve the stress concentration of surface ledge (see Supplementary Fig. 11). The increasing critical strain for dislocation nucleation, successive locking of dislocation nucleation sites, and relieved stress concentration by hydrogen charging clearly indicate that hydrogen atoms can suppress dislocation nucleation at surface ledges.

To quantify the activation energy changes associated with surface dislocation nucleation in the presence of hydrogen, a modified nudged elastic band method (NEB)[26,53–56] was applied to the case shown in Fig. 4a (hydrogen free) and the case in Fig. 4d (all of the surface ledges were charged with hydrogen); see the Methods section for details of the simulation. Figure 4f shows the calculated activation energies of the two cases; the solid lines represent fitting curves by taking the functional form $Q_0(\sigma) = A(1-\sigma/\sigma_{ath})^\alpha$, where $A = 31.3$ eV, $\alpha = 1.46$, $\sigma_{ath} = 2.8$ GPa for the hydrogen-free case and $A = 99$ eV, $\alpha = 1.44$, $\sigma_{ath} = 3.3$ GPa for case d. The NEB results indicate that the activation energy curve of the hydrogen-charged case is significantly shifted rightward. The activation energy for dislocation nucleation at a given stress level increases significantly in the presence of hydrogen; e.g., at a tensile stress of 2.8 GPa, the activation energy increases by at least 5 eV. In cases b–d, hydrogen atoms were inserted right next to the ledges and occupied all possible interstitial sites in the periodic out-of-plane direction. Additional simulations found that when hydrogen atoms are within 2 nm of surface ledges and even partially occupied the interstitial sites, they can effectively suppress dislocation nucleation (see Supplementary Figs. 12, 13). Simulations of stress relaxation and dislocation nucleation with surface roughness/steps further reveal that hydrogen can

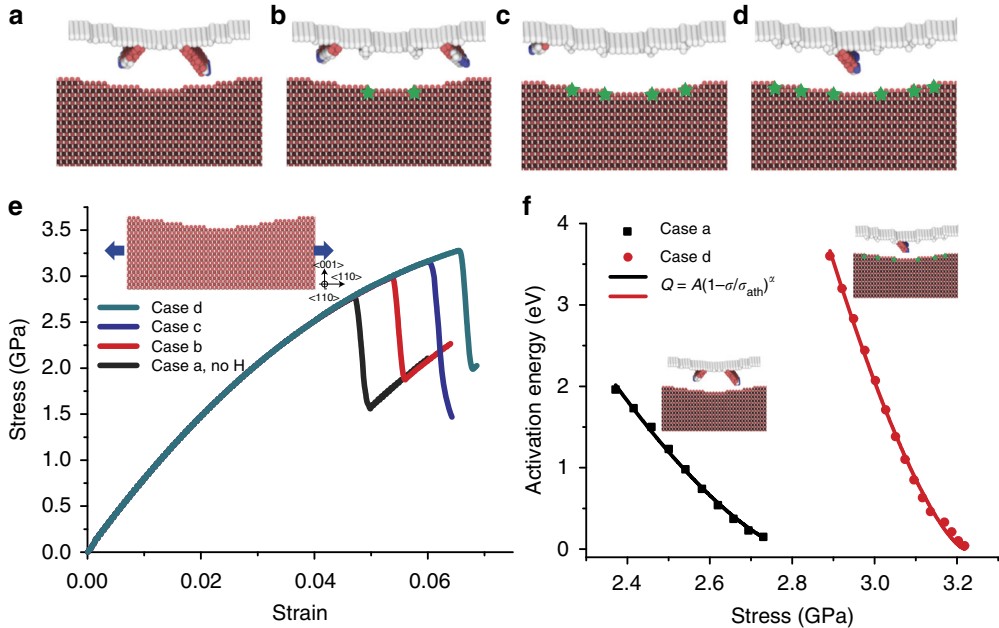

**Fig. 4** MD simulations of dislocation nucleation. **a–d** Upper part of the figures shows dislocation nucleation sites. Green stars in the lower part of the figures indicate positions of hydrogen atoms. **e** Stress–strain curves corresponding to different configurations in **a** to **d**. **f** Stress-dependent activation energy of dislocation nucleation. Black squares correspond to case a without hydrogen; red dots correspond to case d. Solid lines are fitting curves

increase the activation energy of surface dislocation nucleation and thus suppress the surface dislocation nucleation in the NWs. Therefore, the presence of hydrogen can cause fundamental changes in the mechanical properties of metallic NWs, resulting in enhanced yield strength and localized necking, as observed in experiments.

## Discussion
Our in-situ relaxation experiments and simulations have shown that surface-adsorbed hydrogen can suppress surface dislocation nucleation in the NWs; however, a uniform increment of the resistance to surface nucleation is by itself insufficient to bring about the observed localization of plasticity shown in Fig. 1. There must be some non-uniform interactions between surface dislocation and hydrogen. This non-uniformity may exist during the initiation of surface nucleation and the succeeding nucleation.

First, the non-uniformity effect of hydrogen could contribute to the localization of plasticity from the point of view of reduced local nucleation barrier as a hydrogen-locked site breaks free. In an elegant explanation of distributed plasticity in penta-twinned NWs[24], a dislocation nucleation event increases the local activation energy and immediately causes its nucleation site to be locked up, and further nucleation is shifted to a neighboring site where activation energy has not been raised. In the presence of hydrogen, this situation is different because hydrogen has caused a substantial increase in local activation energies. One can imagine that after a dislocation nucleation event, one source is activated and, as soon as it is activated, the local activation energy drops because this site has been released from the constraint of hydrogen while all the other sites are still inhibited by hydrogen. This would then cause an avalanche effect in that the whole system fails at this weakened point. To support this hypothesis, activation energy changes associated with successive dislocation nucleation events were calculated based on the surface ledge model in Supplementary Note 7. It was found that, following an initial nucleation event, activation energy of local nucleation decreases in the presence of hydrogen, while activation energy of non-local nucleation increased significantly due to the effect of

hydrogen. The presence of hydrogen substantially increases $\Delta E$ between non-local nucleation and local nucleation, as shown in Supplementary Fig. 14b. This clearly promotes localized plasticity.

Furthermore, the presence of hydrogen could promote localized plasticity in another way that involves non-uniform hydrogen distribution. Before initial nucleation, we can assume that the activation energy at surface nucleation sites follows a normal distribution. The presence of hydrogen could increase both the mean value and the variance of the activation energy, inducing non-uniformities in the process. Even if hydrogen was initially evenly distributed in the system, the diffusion and interaction between hydrogen atoms and potential nucleation sites are non-uniform, resulting in an increased variance of the activation energy for all nucleation sites. During the initial stage of nucleation, localization occurs if after the first dislocation nucleation, there is no nucleation at all the other sites within a δt time interval; in contrast, distributed plasticity occurs if nucleation readily occurs at other sites following the first dislocation nucleation within a δt time interval. The probability of localization can thus be modeled as a non-homogeneous Poisson process, with upper bound depending on the variance of the activation energy as

$$P_{\text{localization}} = \left(\max_{s'} \lambda(s', \tau_0)\right)\delta t \exp\left\{-\sum_{S} \lambda(s, \tau_0)\delta t\right\} \quad (1)$$

where $s$ denotes a nucleation site and $\lambda = \nu_0 \exp\left\{-\frac{Q}{kT}\right\}$ the nucleation rate. Further details of the model can be found in Supplementary Note 8. If $\lambda$ is assumed to follow a log normal distribution, $\max_{s'} \lambda(s', \tau_0)$ becomes the critical term in (1) and $P_{\text{localization}}$ would increase with an enhanced variance in activation energy. MD simulations on a surface ledge model with different distribution of hydrogen are discussed in Supplementary Note 9. This result clearly shows that at the initiation stage of surface dislocation nucleation, the increment in the variance of the activation energy due to non-uniform interaction between hydrogen and the nucleation sites can lead to more localized nucleation.

The effect of hydrogen in NWs with large surface-to-volume ratios is distinct from traditional HE in bulk materials, where HELP was observed by in-situ TEM. The origin of this is that small-scale NWs are in a dislocation-starved state, where deformation mechanism is governed by dislocation nucleation, whereas in a bulk material, the effect of hydrogen is mostly manifested through its interaction with the movement and multiplication of pre-existing dislocations. The observed hardening in NWs is also in contrast to softening in pop-in stress found in indentation tests[27]. Although the exact reason of such softening has not been satisfactorily explained, our speculation is that the interaction of hydrogen with bulk dislocation sources underneath an indenter or with pre-existing dislocations may have played a significant role during indentation. NW surface defects such as roughness and steps are known to have relatively high binding energy to hydrogen atoms[3,57]. For hydrogen atoms diffusing to and trapped at these sites, desorption is less likely to occur when compared with a perfect surface. The suppression of surface dislocation nucleation as observed in experiments can be attributed to those hydrogen atoms that remain adsorbed and trapped at the defect sites on the NW surface. In our present study, we have clearly observed a hardening effect of hydrogen associated with the surface nucleation in NWs. These findings could have broad implications on the effect of hydrogen on the deformation and failure of nanostructures with large surface-to-volume ratios.

In summary, we have used metallic NWs as a new platform to study HE and discovered a transition from distributed plasticity to localized necking by an integrated approach that combines in-situ TEM tensile testing, microstructure characterization, and MD simulations. The effect of hydrogen on mechanical behavior of nanostructures manifested itself dominantly through dislocation nucleation. The ultimate strength of the NWs increased significantly in the presence of higher hydrogen concentration. In-situ TEM relaxation tests and simulations further showed that the stress relaxation behavior in penta-twinned Ag NWs governed by dislocation nucleation was attenuated as the hydrogen concentration increased. These results together suggested that hydrogen can effectively suppress surface dislocation nucleation in metallic NWs. NEB simulations found that hydrogen increases the activation energy of dislocation nucleation from single-atom surface ledges. Therefore, in contrast to previous studies in bulk materials showing the effect of hydrogen on dislocation movement and multiplication, the present study revealed that hydrogen can fundamentally affect the nucleation-controlled deformation and failure mechanisms in nanostructures. By investigating HE of a clean material system such as NWs, the present study can shed valuable insights into mechanisms of HE in bulk materials.

## Methods

**Sample preparation and characterization**. Penta-twinned Ag NWs were synthesized by reducing $AgNO_3$ with ethylene glycol in the presence of polyvinyl pyrrolidone. The solution of Ag NWs was diluted with deionized water and then purified by centrifugation. More details of the NW synthesis process are provided elsewhere[38].

For hydrogen charging, Si substrates containing Ag NWs from the same synthesis batch were used to ensure the crystalline quality and dimensions of the tested NWs. Before hydrogen charging, the substrate containing Ag NWs was purged for 2 h in a vacuum chamber with a constant flow of pure Ar (99.99999%) at 250 °C. Then, they were immerged into a constant flow of $H_2$/Ar (molar ratio, 1:1) for hydrogen charging at room temperature. Time-of-flight SIMS was used to examine the relative concentration of hydrogen inside the samples with different charging times (see details in Supplementary Fig. 1 and Supplementary Table 1).

**In-situ TEM mechanical testing**. Mechanical testing was carried out in situ inside a TEM using a MEMS-based nanomechanical testing stage, which consists of a thermal actuator, a capacitive load sensor, and a gap in between for mounting

samples (Supplementary Fig. 4a). Details on the testing stage have been previously reported[46,58]. Displacement (and strain) is measured by digital image correlation of TEM images of two local markers on the specimen (Supplementary Fig. 4b). The MEMS-based testing stage has a strain resolution of 0.01% (gauge length 2 μm) and a stress resolution of 1.4 MPa (e.g., for NWs with a diameter of 104 nm[50]).

NWs were mounted on the testing stage using a nanomanipulator (Klocke Nanotechnik, Germany) inside an FEI Nova 600 dual beam[50,58]. Two local markers were deposited on the NW for displacement (and strain) measurement. In-situ TEM mechanical testing was performed on JEOL 2010F operated at 200 kV. The loading and unloading strain rates for in-situ TEM tensile testing were ~0.005%/s. During loading, the actuator was loaded incrementally to a prescribed displacement. During relaxation, the actuator displacement was held constant and the specimen relaxed as a function of time. The load on the specimen decreased and the specimen elongation increased at the same time. During recovery, the actuator was turned off and retracted to the original position.

Low-magnification images were recorded at a fixed condense (the second condense lens) current to minimize the focus change. The current density of the incident e-beam is <0.1 A/cm² and its effect on the mechanical behavior of the NW under tensile testing can be ignored.

**Atomistic simulations**. MD simulations were performed using the software package LAMMPS[56] and the atomic configurations were displayed by OVITO[59]. Atoms in Figs. 2, 3 are colored according to their crystalline structure: green for face-centered cubic symmetry, red for hexagonal close-packed symmetry, and gray for the atoms at dislocation cores, surfaces and point defects. The embedded atom method potential for Ag and H was used to describe the interatomic interactions[60]. *Tensile testing*: The penta-twinned NW samples were 15 nm in diameter and 80 nm in length. Then, 1.2% of Ag atoms were randomly removed from the samples to introduce vacancies. Hydrogen atoms were initially inserted on the free surface of NW samples at random. The samples were initially relaxed and equilibrated at an elevated temperature of 800 K (~65% of the melting point of Ag) for 500 ps using the Nosé-Hoover thermostat. Periodic boundary conditions were imposed along the axial direction. During loading, the samples were stretched at a constant strain rate of $10^6 \, s^{-1}$ under the canonical ensemble (NVT). Five different samples were tested for each hydrogen concentration. *Stress relaxation*: The penta-twinned NW samples were 24 nm in diameter and 30 nm in length. Then, 1.0% of Ag atoms were randomly removed from the samples to introduce vacancies. Hydrogen atoms were initially inserted on the free surface of NW samples at random. The samples were initially relaxed and equilibrated at an elevated temperature of 800 K for 500 ps using the Nosé-Hoover thermostat. Periodic boundary conditions were imposed along the axial direction. The samples were first stretched to a certain stress at a constant strain rate of $10^8$/s and then relaxed for several nanoseconds under NVT ensemble while holding the strain fixed. During the relaxation process, we monitored the variation of the axial stress by averaging the Virial stresses over all atoms in the samples. *Nucleation on ledges and NEB calculation*: The simulation box was oriented along x-[110], y-[001], z-[1̄10], with dimensions of 26 nm × 50 nm × 10 nm. Periodic boundary conditions were applied in the x and z direction. Hydrogen atoms were inserted into octahedral sites in the designated positions (lowest energy sites close to surface step). The system was initially relaxed and equilibrated at temperature of 5 K for 500 ps using the Nosé-Hoover thermostat, and then tensile loading was applied in the x direction at a strain rate of $10^8 \, s^{-1}$ under NVT ensemble. For NEB calculations, the initial states were the system with no dislocation at different stresses. The final states were created by a remap of the system with a dislocation embryo to the corresponding strain and minimized so that the system energy is only 0.01 eV lower than the corresponding initial replica. During NEB, constraints were applied on last replica and set its target energy to that of first replica[55].

## Data availability

The data that support the findings of this study are available from the corresponding author upon reasonable request.

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

## Acknowledgements

We acknowledge financial support from the National Science Foundation (NSF) under Award Numbers DMR-1410475 and DMR-1709318. We acknowledge the use of the Analytical Instrumentation Facility (AIF) at North Carolina State University, which is supported by the State of North Carolina and NSF (Award Number ECCS-1542015) and computational support by the Extreme Science and Engineering Discovery Environment (XSEDE) through Grant MS090046.

## Author contributions

S.Y. and G.C. contributed equally to this work. Y.Z. and H.G designed the project and guided the research. G.R. synthesized the NWs. G.C. and T.H.C. conducted the in-situ TEM experiments. S.Y. performed the atomic simulations and statistical analysis. S.Y., G.C., Y.Z. and H.G. co-wrote the paper.

## Additional information

**Competing interests:** : The authors declare no competing interests.

**Journal Peer Review Information**:*Nature Communications* thanks the anonymous reviewers for their contribution to the peer review of this work. Peer reviewer reports are available.

