## [Peer Review File · Nature Communications]

Reviewers' comments:

Reviewer #1 (Remarks to the Author):

This paper reports the in situ TEM testing and atomistic modeling of H effects on the mechanical behavior of penta-twinned Ag nanowires (NWs). The in situ TEM results show the increased yield strength and a transition from distributed plasticity to localized necking in Ag NWs with increasing H concentration. In situ relaxation experiments and atomistic simulations provide insight into the H effects on surface dislocation nucleation that may control the yielding responses of Ag NWs.

This work represents an important progress in utilizing the in situ quantitative nanomechanical tests for studying a long-standing problem of hydrogen embrittlement of metallic materials. The results are very interesting, as they contrast with the previous in situ TEM observation of hydrogen enhanced local plasticity – the so-called HELP mechanism. Overall, the paper is clearly written. But there are several critical details that should be clarified.

The authors used SIMS to determine the H concentration in Ag NWs that can be as high as $\sim 1\%$. The H concentration in metals is conventionally given in terms appm or wt ppm. What is the exact meaning of $\sim 1\%$ H concentration (appm or occupancy of interstitial sites)? More importantly, given the H charging at room temperature, the H concentration of 1% is very high, compared to the H solubility in most FCC metals reported in the literature. More careful benchmark experiments are needed to substantiate the reported values of H concentrations.

There are two types of interstitial sites (i.e., octahedral and tetrahedral sites) for H trapping in an FCC lattice. The two sites have different binding energies with H atoms, which determine the respective equilibrium H concentration. There are also similar site competition issues for H trapping at the surface. What kind of bulk and surface binding sites are studied for trapping H atoms in this work? Why?

The CINEB modeling of surface dislocation nucleation requires some clarification. For the CINEB method, the final state should be a local energy minimum that is fully relaxed without any constraint. The authors fixed the final image/replica with a nucleated surface dislocation at the energy of 0.01eV lower than that of the initial state. It seems to be an incorrect operation of the CINEB method, as the final state should not be fixed at a particular energy level. Moreover, at different load levels, the final state must have different energy values relative to the corresponding initial state. So it would be impossible to fix the final image at a particular energy value relative to the initial state.

In MD simulations, it is not clear why the presence of H can give rise to a transition from distributed plasticity to localized necking. Is it due to the non-uniform H distribution in MD simulations? In other words, the uniformly disturbed H could increase the resistance to surface dislocation nucleation, but may still lead to distributed dislocation plasticity similar to the case without H. The cause of the transition from distributed plasticity to localized necking in both experiments and MD should be clearly discussed.

This work contrasts with the previous in situ TEM observation of hydrogen enhanced local plasticity – the so-called HELP mechanism. This difference and possible causes should be discussed.

Other minor issues:

Since the CINEB calculations are performed for a quasi-2D setup, the calculated energy barrier should be given in terms of eV/Å.

It is suggested to denote the H concentration as c_H instead of Δ_H , since the latter is often

used to represent the H binding energy/enthalpy.

In line 119, the unit of strain rate of 10^6 is missing.

Reviewer #2 (Remarks to the Author):

See attached file.

Reviewer #3 (Remarks to the Author):

Dear authors,

You have done good experimental work and combine it perfectly with the simulation to achieve very good results. I enjoyed the reading as it gives new interesting insight on the mechanism of hydrogen embrittlement. Unfortunately, there are some minor mistakes which need in some work. See the following comments.

Comment: Figure 1a: Give scale bar length in drawing and not only in the figure caption

Comment: Figure 1b: Give the strain rate of the tests also in figure caption.

Comment: Figure 1c and 1d: Explain the numbering "i" to "vi" in the figure caption or directly on the picture. You explain it in the text from line 96 to line 101 but it is not understandable, when looking solely on the figure. Furthermore, in the text you explain that "I" to "iv" represent different load levels (or strains). Could you also give the absolute values of load or strain which corresponds to single pictures i to iv.

Comment: Figure 1c and 1d: Give the size of the scale bar in the figure.

Comment: Supplementary figures and tabular. I received the supplementary information and the paper in two different files. I suggest to combine those and to add the supplementary information as appendix in the main paper file for publishing. This helps the reader not to oversee any information when downloading the file once published, especially as you give very helpful information in the supplement to answer arising questions, e.g. the hydrogen content. I am aware that you have to check with the politics of the magazine.

Comment: Hydrogen concentration after charging: The normalized hydrogen concentration is given with the uncharged sample as reference value. Could the absolute (ppm) hydrogen concentration be given?

Comment: Concerns Fig 2a: Can you give a scale bar? What are the dark and green areas? Am I right that dark areas are distributed plasticity and green areas are without plasticity.

Comment: Fig 2b: Is it possible to use different lines styles, e.g. full line or dotted line, to allow a color-blind reader to distinguish between the lines.

Comment: Fig 3c: I took me some time to realize that the dark areas on the penta-twinned NWs represent dislocations. Please add this information in the figure caption. Furthermore, the legend gives the hydrogen concentration with no hydrogen on top and maximum hydrogen at the bottom. It would be more logical for me to sort the legend in the order of the stress-relaxation-lines in the figure.

Comment: Fig 4e: Similar to the last comment. I would sort the legend according to the lines. Case d on top as it is the curve on top.

Comment: Line 91: You say that the NWs vary in diameter by about $\pm 4\%$. Is there any effect of this variation of the cross section on the elastic-plastic behavior the NWs? You may add a small comment on this in the paper.

Comment: Line 91-92: You refer to the "Methods". Help the reader to find the "methods"-chapter. You may say that there is an appendix.

Comment: Line 99: You write that necking is observed in the NWs. Can you give the reduction of area?

Comment: Line 121: You write "Figure 3 shows snapshots of the observed failure modes". Are you sure you mean figure 3 and not 1b. If you mean figure 3 you should explain in the text what to see in figure 3 when referring to it as it is used here first time in the text.

Comment: You give the hydrogen concentration only in the supplements. I think this information is important to rate the results. I suggest adding this information to the main part of the paper.

Comment: Line 125m, Chapter about the tensile test. Are you sure you want to refer to Figure 3b? For me it is more logical to see the higher yield stress in the stress-strain curves in Figure 1b?

Comment: Concerns the tensile test. You do not give the strain rate in the main part of the paper but only in the methods chapter. For bulk material tests the strain rate can – especially when there is HE- have a big effect on the fracture strain. Mostly the lower the strain rate, the higher the HE and the lower the strain at fracture. Compare standard ASTM G 129 which considers this behavior in an international standard. Also the chosen strain rate of $5e-5$ 1/s is, at least for some bulk material, a rather high strain rate, when looking on the effect of HE in mechanical test. Could you explain why you have chosen this strain rate? Did you do test with other strain rates to see the strain rate effect?

Comment: Concerns the hydrogen desorption. This comment might be linked to the answer of the last comment concerning the strain rate. I understand that you have done the tensile tests and the relaxation test on hydrogen pre-charged specimen. Normally, at least in bulk steel, after hydrogen pre-charging, the hydrogen desorption/effusion starts directly after the end of charging. Is there hydrogen effusion in gold NWs? When there is effusion, what is the effusion rate? How much hydrogen desorbs between the end of charging and the start of the test. I suspect that the tests in the TEM are done in vacuum and this should support the hydrogen effusion. How much hydrogen remains in the NWs after the end of the mechanical tests? I expect some proof that there is hydrogen in the NWs when testing

Comment: Line 148 and Fig. 3a: Can you give the strain rate and the hold time at maximum and minimum load. In the text (line 148) you name the test "tensile test" and in the figure caption (Fig. 3a) "relaxation test". Those are relaxation tests (or hysteresis loops). Use the same name throughout the paper.

Comment: Chapter MD simulations of dislocation nucleation and activation energy, starting line 165: You give simulation evidence and experimental results that "hydrogen can suppress surface dislocation nucleation (authors cited in line 166-167)" in metallic nano wires.

I miss direct experimental evidence in the paper. I expect some TEM pictures of dislocations structures to proof this. Would it be possible to make TEM lamellas with FIB from the tested NWs with different hydrogen concentration to analyze the dislocation structures? You could compare the dislocations structures of the different NWs in the area of necking and outside the necking area.

Then you would have direct evidence that the hydrogen reduces the dislocation nucleation. Alternatively you could cite references in which this mechanism is shown.

Kind regards

Response to Reviewers' comments:

We would like to thank the reviewers for their constructive comments that have helped us improve the quality of the manuscript. We have addressed all the comments/questions raised by the reviewers. Please find attached the revised manuscript, along with a **List of Changes** and detailed point-to-point **Response to the Reviewer Comments**. The revised texts have been highlighted in red in the revised manuscript.

List of changes:

1. Page 3, line 11 was revised as “and no “shielding” of dislocation–dislocation interactions by hydrogen can be found in atomic simulations¹⁶. Hydrogen segregation to grain boundaries and interactions with vacancies also play important role in hydrogen embrittlement¹⁷⁻¹⁹. A recent study suggested that vacancies can interact with hydrogen and lock dislocations motion in aluminum²⁰. A chemomechanical origin of hydrogen trapping and segregation at GBs was revealed in recent atomic level studies of hydrogen embrittlement^{21,22}.”.
2. Page 3, line 21 was revised as “In bulk materials, softening in pop-in stress during nanoindentation in the presence of hydrogen has not yet been clearly understood. Atomistic simulations revealed that hydrogen-induced swelling of the lattice²⁸ can contribute to the softening but only to a limited extent.”.
3. Page 5, line 10 was revised as “14 to 75 wt ppm”.
4. Page 5, line 19, “It has been reported that the mechanical properties of penta-twinned Ag nanowires are strongly size dependent^{24,41}. To focus our attention on the effect of hydrogen,” was inserted.
5. Page 6, line 1 was revised as “ c_H from 0 to 75 wt ppm”.
6. Page 6, line 9 was revised as “ $c_H = 75$ wt ppm”.
7. Page 6, line 12 was revised as “ $c_H = 38$ wt ppm”.
8. Page 7, line 6, “/s” was inserted.
9. Page 7, line 8, “Supplementary Figs. 4-5” was inserted.
10. Page 7, line 8 was revised as “Figure 2a shows ...”.
11. Page 7, line 12 was revised as “Fig. 2b show ...”.
12. Page 8, line 1, “Specifically, we found that the drop in local activation energy associated with sudden release of a hydrogen-locked site and the interaction between nonuniform hydrogen distribution and surface nucleation sites promote localized plasticity; see Discussion section for details.” was inserted.
13. Page 8, line 7 was revised as “stress relaxation behavior”, “and strain recovery” was deleted.
14. Page 8, line 14, “Thus recoverable plasticity⁴⁵ was observed in penta-twinned NWs in the presence of hydrogen, and the change in the magnitude of stress relaxation at different hydrogen concentrations reveals an essential role of hydrogen on surface nucleation.” was inserted.
15. Page 8, line 17, “tensile tests” was replaced by “relaxation tests”.
16. Page 9, line 16, “(0 to 94 wt ppm)” was inserted.
17. Page 10, line 3, “57 wt ppm” was inserted.
18. Page 11, line 19, “a modified nudged elastic band method” was inserted.

19. Page 12-Page 15, a new **Discussion** part was inserted.
20. Page 18, line 23, “due to the binding energy difference with tetrahedral sites” was inserted.
21. Page 19, line 18, “(lowest energy sites close to surface step)” was inserted.
22. Page 20, line 1, “During NEB, constraints were applied on last replica and set its target energy to that of first replica” was inserted.
23. Page 20, line 4, “**Data availability ...**” was inserted.
24. Page 24, figure 1, “i to iv correspond to strains of 0, 1.3, 1.4 and 4.9%.” was inserted. “i to iv correspond to strains of 0, 1.5, 2.0 and 2.3%.” was inserted. “The strain rate for *in situ* TEM tensile testing was ~0.005%/s.” was inserted.
25. Page 25, figure 2b was replaced by curves with error bar. The caption was revised as “Averaged stress-strain curves of Ag NWs at different hydrogen concentrations with error bars.”.
26. Page 26, figure 3(a), legend **relaxation** was bolded. “Red atoms in the grain show stacking faults in the right insets.” was inserted in the figure caption.
27. Some new references were included.

In Supplementary Information:

1. Page 3, wt ppm column was inserted in Supplementary Table 1.
2. Page 5, “i to iv correspond to the strains of 0, 1.5, 1.8 and 2.7%.” was inserted in caption.
3. Page 5, “In Supplementary Fig 5, we investigated the 300K case at higher strain rate of $10^8/s$. Comparing with Supplementary Fig 4b, the difference is obvious. The presence of hydrogen promotes dislocation nucleation in this case due to the limited hydrogen diffusion at higher strain rate. These results further stress that the diffusion and interaction time scale of hydrogen is of critical importance in the simulations of surface nucleation.” was inserted.
4. Page 8, new figure Supplementary Figure 5 was inserted.
5. Page 12, new discussion part: “Activation energy change associated with successive dislocation nucleation” was inserted with new Supplementary Figure 9.
6. Page 15, new discussion part: “S7. Localization via a nonhomogeneous Poisson process” was inserted.
7. Page 18, Supplementary Figure 10 was inserted.
8. Page 17, new discussion part: “S8. Localization due to variance of hydrogen in the surface ledge model” was inserted with new Supplementary Figure 11.

Response to the Reviewer Comments:

Reviewer #1 (Remarks to the Author):

This paper reports the in situ TEM testing and atomistic modeling of H effects on the mechanical behavior of penta-twinned Ag nanowires (NWs). The in situ TEM results show the increased yield strength and a transition from distributed plasticity to localized necking in Ag NWs with increasing H concentration. In situ relaxation experiments and atomistic simulations provide insight into the H effects on surface dislocation nucleation that may control the yielding responses of Ag NWs.

This work represents an important progress in utilizing the in situ quantitative nanomechanical tests for studying a long-standing problem of hydrogen embrittlement of metallic materials. The results are very interesting, as they contrast with the previous in situ TEM observation of hydrogen enhanced local plasticity – the so-called HELP mechanism. Overall, the paper is clearly written. But there are several critical details that should be clarified.

Response:

We thank the reviewer for her/his positive assessment of our paper.

1. The authors used SIMS to determine the H concentration in Ag NWs that can be as high as ~1%. The H concentration in metals is conventionally given in terms appm or wt ppm. What is the exact meaning of ~1% H concentration (appm or occupancy of interstitial sites)? More importantly, given the H charging at room temperature, the H concentration of 1% is very high, compared to the H solubility in most FCC metals reported in the literature. More careful benchmark experiments are needed to substantiate the reported values of H concentrations.

Our Response:

Based on our knowledge, the Time-of-Flight Secondary Ion Mass Spectrometer (TOF-SIMS) is the best way to estimate the concentration of hydrogen introduced during the charging process. We calculated the hydrogen concentration, $H/(H+Ag)$, from the SIMS results. In response to the reviewer's comments, we have changed the unit of hydrogen concentration to wt ppm (1 wt ppm = 1 mg/kg).

The hydrogen absorbed on the surface of the NWs can diffuse into the Ag NWs (as interstitials) during the charging process under a H₂/Ar (molar ratio, 1:1) atmosphere, owing to the high diffusivity ($\sim 10^{-14}$ m²/s) of hydrogen. Here the high hydrogen concentration can be attributed to the large surface-to-volume ratio and the small diameter (<100 nm) of the NWs.

In the manuscript, we have changed the atomic percentage of hydrogen to wt ppm.

Charging time	c_H	
	at.%	wt ppm
hrs		
0	0	0
12	0.15	14
24	0.41	38
36	0.63	59
48	0.8	75

Supplementary Table 1 | Hydrogen difference (c_H) in Ag NWs with different charging time into a H₂/Ar (molar ratio, 1:1) atmosphere, normalized to the as-received samples (not immersed in H₂/Ar atmosphere).

2. There are two types of interstitial sites (i.e., octahedral and tetrahedral sites) for H trapping in an FCC lattice. The two sites have different binding energies with H atoms, which determine the respective equilibrium H concentration. There are also similar site competition issues for H trapping at the surface. What kind of bulk and surface binding sites are studied for trapping H atoms in this work? Why?

Our response:

In our simulations, the hydrogen atoms were randomly inserted in the octahedral interstitial sites of FCC lattice, since octahedral interstitial has a lower binding energy than the tetrahedral interstitial (0.53 eV energy difference in our MD simulation). This is consistent with the common belief that H occupies octahedral sites in most fcc metals as shown in the literature [1-2].

In addition, we have also calculated the migration energy barrier of hydrogen between nearest octahedral and tetrahedral sites through the NEB method. The result shows that tetrahedral to octahedral has the lowest barrier for migration (0.0154 eV). The energy barriers of other migration paths are: 0.55 eV from octahedral to tetrahedral, 0.098 eV from tetrahedral to tetrahedral and 0.62 eV from octahedral to octahedral. So in our simulations, initially all the hydrogen atoms were randomly inserted in the octahedral interstitial sites. Besides, in our tensile simulation, we used high temperature (800 K) and extremely low strain rate (10^6 /s) to ensure that hydrogen has enough time to diffuse.

In the surface ledge model, hydrogen atoms were directly inserted at the octahedral interstitial sites next to the surface step, which leads to the lowest potential energy for the whole system as shown in Figure R1.

Figure R1. Hydrogen sites around the surface step model. a. The atomic structure around a surface step, viewed from $\langle 110 \rangle$ direction. b. Potential energy contour for different hydrogen interstitial sites (all octahedral sites). The dark blue one is the position with lowest total potential energy after hydrogen insertion.

We have added this information in the method part, and included a discussion about the GB embrittlement behavior in the introduction.

[1]. Zhou, X., et al. "Chemomechanical origin of hydrogen trapping at grain boundaries in fcc metals." *Physical review letters* 116.7 (2016): 075502.

[2] Bugeat, J. P., and Ligeon, E.. "Lattice location and trapping of hydrogen implanted in FCC metals." *Physics Letters A* 71.1 (1979): 93-96.

3. The CINEB modeling of surface dislocation nucleation requires some clarification. For the CINEB method, the final state should be a local energy minimum that is fully relaxed without any constraint. The authors fixed the final image/replica with a nucleated surface dislocation at the energy of 0.01eV lower than that of the initial state. It seems to be an incorrect operation of the CINEB method, as the final state should not be fixed at a particular energy level. Moreover, at different load levels, the final state must have different energy values relative to the corresponding initial state. So it would be impossible to fix the final image at a particular energy value relative to the initial state.

Our Response:

The NEB method adopted here is not the original CINEB method, rather it is similar to the free-end NEB method in the literature [1,2]. Usually, the sample without dislocation is in a local minimum state; however, the sample with dislocation embryo is not at a local minimum energy level in most cases, and an extra constraint need to be applied to the final replica. In the latest manual of LAMMPS, `fix_neb` method has the keyword "*last/efirst*" which will apply a force to the last replica and set its target energy to that of the first replica [3]. This keyword was adopted in our calculation.

In our calculation, we first minimized the initial state without dislocation embryo to the energy minimum state and use it as the first replica. When the sample with dislocation embryo was mapped to the same strain as the first replica, the energy state was different from the first replica, as the reviewer pointed out. Then we need to do an energy minimization of the sample with dislocation embryo at the same total strain until its energy was very close to the initial replica (<0.01 eV), and we terminated the minimization and use this configuration as the last replica. Finally, the keyword "last/efirst" was used to fix the energy of last replica during the NEB calculation in lammmps.

We have revised the method part, added more detail of simulation procedures and included the following references.

[1] Zhu, T., et al. "Interfacial plasticity governs strain rate sensitivity and ductility in nanostructured metals." *Proceedings of the National Academy of Sciences* 104.9 (2007): 3031-3036.

[2] Zhu, T., et al. "Temperature and strain-rate dependence of surface dislocation nucleation." *Physical Review Letters* 100.2 (2008): 025502.

[3] Maras, E., et al. "Global transition path search for dislocation formation in Ge on Si (001)." *Computer Physics Communications* 205 (2016): 13-21.

4. In MD simulations, it is not clear why the presence of H can give rise to a transition from distributed plasticity to localized necking. Is it due to the non-uniform H distribution in MD simulations? In other words, the uniformly disturbed H could increase the resistance to surface dislocation nucleation, but may still lead to distributed dislocation plasticity similar to the case without H. The cause of the transition from distributed plasticity to localized necking in both experiments and MD should be clearly discussed.

Our Response:

This is a very good question that deserves a dedicative investigation. As the reviewer pointed out, only increment of the resistance to surface nucleation due to hydrogen is insufficient to induce localization of plasticity. There must be some non-uniform interactions between surface dislocation and hydrogen. The nonuniformity may exist in the initiation of surface nucleation stage, and/or during the succeeding nucleation stage.

The nonuniformity of hydrogen effect could contribute to the localization during the succeeding nucleation stage. In an elegant explanation of distributed plasticity in penta-twinned NWs [1], a dislocation nucleation event increases the local activation energy and immediately causes its nucleation site to be locked up, and further nucleation is shifted to a neighbouring site where activation energy has not been raised. In the presence of hydrogen, this situation could be different because hydrogen has caused a substantial increase in local activation energies. One can imagine that after a dislocation nucleation event, one source is activated and as soon as it is activated, the local activation energy drops because this site has been released from the constraint of hydrogen while all the other sites are still inhibited by hydrogen. This would then cause an avalanche effect in that the whole system fails at this weakened point. To support this

hypothesis, the activation energy changes associated with succeeding dislocation nucleation were calculated based on the surface ledge model in Supplementary Material S-6. It was found that activation energy of local nucleation even decreases a bit in the presence of hydrogen, while activation energy of non-local nucleation increased significantly due to the effect of hydrogen. Overall, the presence of hydrogen substantially increases ΔE between non-local nucleation and local nucleation as shown in Supplementary Figure 9(b) and thus will clearly promote localized plasticity.

Furthermore, the presence of hydrogen promotes localized plasticity in another way that involves non-uniform hydrogen distribution. For the nonuniformity during the initiation of surface dislocation, we can assume that the activation energies at the surface nucleation sites follow a normal distribution during loading. The presence of hydrogen could, in addition to obviously increasing the mean value of the activation energies, increase the variance of the activation energies. The nonuniformity originates from the increment in the variance of the activation energies. Although the hydrogen is evenly distributed in the system initially, the diffusion and interaction between hydrogen atoms and nucleation sites are non-uniform. Supplementary Figure S10 shows the distribution of hydrogen on NW surface in MD simulation. While it does not show the activation energy distribution on the nucleation sites, non-uniform distribution of hydrogen on NW surface can be observed. During initial nucleation, localization occurs if after the first dislocation nucleation, there is no nucleation at other sites within a δt time interval; in contrast, distributed plasticity occurs if after the first dislocation nucleation, within a δt time interval, there is nucleation at other sites. The probability of localization can thus be modeled as a nonhomogeneous Poisson process, with upper bound depending on the variance of the activation energy as

$$P_{localization} = (\max_{s'} \lambda(s', \tau_0)) \delta t \exp\{-\sum_S \lambda(s, \tau_0) \delta t\}$$

where s denotes a nucleation site and $\lambda = \nu_0 \exp\{-\frac{Q}{kT}\}$ the nucleation rate. Further details of the model are provided in Supplementary Material S-7. If λ is assumed to follow a log normal distribution, $\max_{s'} \lambda(s', \tau_0)$ becomes the critical term in (1) and $P_{localization}$ would increase with an enhanced variance in activation energy. MD simulations on a surface ledge model with different distribution of hydrogen were carried out in Supplementary Material S-8. This result clearly shows that at the initiation stage of the surface dislocation nucleation, the increment in the variance of the activation energies due to non-uniform interaction between hydrogen and the nucleation sites can lead to more localized nucleation.

Supplementary Figure 9 | NEB calculation of the nucleation activation energy change. (a) Two different nucleation configurations in the surface ledge model. (i), local nucleation: new partial nucleated at the extrinsic stacking fault created by the initial nucleation (blue dashed circle). (ii), non-local nucleation: new partial nucleated (blue dashed circle) at a surface ledge far away from the initial nucleation site where the extrinsic stacking fault was left. (b) Activation energy of two different scenarios with/without hydrogen as function of tensile strain.

Supplementary Figure 11 | Localization in the surface ledge model. (a) Hydrogen distribution in twelve surface ledges. The total number of hydrogen atoms are fixed, but the variance of concentration of hydrogen on surface ledges keeps increasing from (i)-(iv). (b) Snapshots of nucleation with different distributions of hydrogen from initial nucleation to 12ps after initial nucleation (extra 0.12% strain). The nucleation sites were marked by blue circles.

We have added this in the manuscript Discussion part.

New NEB calculation of activation energies of succeeding dislocation nucleation was added in SI Section S6.

Also detail of the statistical modeling is included in the SI Section S7.

New MD simulations on surface ledge model that reveal the transition was added in SI Section S8.

[1] Filleter, T., et al. "Nucleation-controlled distributed plasticity in penta-twinned silver nanowires." *Small* 8.19 (2012): 2986-2993.

5. This work contrasts with the previous in situ TEM observation of hydrogen enhanced local plasticity – the so-called HELP mechanism. This difference and possible causes should be discussed.

Our Response:

Thanks for the suggestion; we have added a paragraph of discussion before the conclusion.

“The effect of hydrogen in NWs with large surface-to-volume ratios is distinct from traditional hydrogen embrittlement in bulk materials, where hydrogen enhanced local plasticity (HELP) was observed by in-situ TEM. The origin of this is that small scale NWs are in a dislocation starved state, where deformation mechanism is governed by dislocation nucleation while in a bulk material, the effect of hydrogen is mostly manifested through its interaction with the movement and multiplication of pre-existing dislocations.”

Other minor issues:

6. Since the CINEB calculations are performed for a quasi-2D setup, the calculated energy barrier should be given in terms of eV/Å.

Our Response:

The thickness in the out-of-plane direction is 5 nm. Although it is periodic in this direction, the nucleated dislocation loop is still a 3D dislocation loop, so the units of energy barrier should still be eV.

7. It is suggested to denote the H concentration as c_H instead of ΔH , since the latter is often used to represent the H binding energy/enthalpy.

Our Response:

Thanks for the suggestion. We have changed ΔH to c_H .

8. In line 119, the unit of strain rate of 10^6 is missing.

Our Response:

Thanks. We have added the unit /s behind of the value.

Reviewer #2 (Remarks to the Author):

This paper presents some new experimental studies of H interactions in a typical fcc metal (Ag), although within the context of a very special penta - twinned nanowire structure. Complementary MD studies are performed to rationalize some of the experimental observations. H embrittlement is a complicated problem that has resisted clear understanding for a long time. All experiments and simulations that can reveal some effects of H on metal deformation are thus valuable for providing systematic information that might guide a detailed understanding of the problem. In that context, the experimental results are nice to see. But the results are not too surprising given the existing knowledge (experiments and theory/simulations) on H effects in fcc metals. The simulations are suitably designed to study the expected mechanism in the experiments, and appear to be cleanly executed. But again the results are not surprising (see below). Furthermore, in the short NatComm format, necessary details and discussion are not possible. The overall work merits publication, after revision guided by some comments below, but probably in a disciplinary journal (e.g. Acta Materialia) where considerable related work on H in metals at the same quality level has been published (both experiments and modeling). The present work does not seem to bring substantially new understanding or knowledge. The limitation to nanowires, where nucleation is the major issue, also does not seem to enable the present study to shed significant light on actual H embrittlement mechanisms that presumably operate in larger - scale materials. The authors show (Figure 1) lower failure strains with H, but this is due to H - delayed nucleation up to stresses at which the material cannot support stable plasticity, and this is not the case in bulk metals, where strengths are lower than bulk). This should not undercut that the work is solid and a good contribution to the field, but it I don't find it suitable for this journal. Here are comments that the authors should probably consider in revising the paper (and lengthening the paper) for submission to another journal:

Response:

We thank the reviewer for her/his comments that our work is solid and a good contribution to the field. We respectfully disagree with the reviewer regarding the new understanding. Our work combining state-of-art in situ experiment and simulation provides the first study of its kind on hydrogen embrittlement in a dislocation starved system, a problem of broad interest to the reader of Nature Communications.

1. The “hydrogen shielding model” does not affect long - range interactions, nor make dislocations easier to move. The speculations on this have been shown to be incorrect, dating back to original modeling of Sofronis and in spite of persistence of the idea in the literature. Recent work by Song et al. (not referenced here) seems to confirm Sofronis' conclusions. Other work by Von Pezold et al. also shows no long - range effects, although the discussion in that paper does not make any clear statements, if I recall.

Our Response:

Thanks for pointing out the reference we missed [1]. We have added this reference in the revised manuscript.

[1] Song, J., and W. A. Curtin. "Mechanisms of hydrogen-enhanced localized plasticity: an atomistic study using α -Fe as a model system." *Acta Materialia* 68 (2014): 61-69.

2. It is unclear whether nanoindentation pop-in experiments are related to actual nucleation. The referenced paper 26 showed that the referenced paper 25 incorrectly interpreted simulations. The referenced paper 29 presents a model where there is a pre-existing loop, around which H atoms can aggregate. This is thus not nucleation as usually envisioned, nor as studied here. Currently there is no real understanding of the pop-in experiments.

Our Response:

We agree with the reviewer that there is no real clear understanding of nanoindentation pop-in experiment currently. We agree with reference 26 (Zhou, *Acta Mater.* 2016) that in the atomic simulation, hydrogen-induced swelling of the lattice may affect the nucleation but only to a limited extent.

However, atomic simulations reported in the literature were mostly conducted in a clean system, without initial defects and under very high loading rate that will limit the diffusion of hydrogen. We believe that in experiments pre-existing dislocations and defects play critical roles in hydrogen embrittlement in bulk systems, which still need further study.

The NW studied here is known to be in a dislocation starved state, so we can exclude the effect of pre-existing dislocations. But initial defects for surface dislocation and hydrogen atom diffusion still need to be included in the simulations. If the simulation strain rate is high or the temperature is low (see SI Section S3, Supplementary Figures 4 and 5), the diffusion of hydrogen will be limited thus its interactions with defects will be limited. Hydrogen will act as sessile inclusion and the lattice distortion will reduce the threshold for dislocation nucleation, which is consistent with reference 26 (Zhou, *Acta Mater.* 2016). Our current simulation results support the experiment phenomenon we observe, and also emphasize the importance of initial defects and time scale in atomistic simulations.

We have modified the introduction of pop-in experiment as:

“In bulk materials, softening in pop-in stress during nanoindentation in the presence of hydrogen [27] has not yet been clearly understood. Atomistic simulations found that hydrogen-induced swelling of the lattice [28] can contribute to the softening but only to a limited extent.”

3. Note that the pop-in experiments, if real nucleation, expressly contradict the current experiments. So, is nucleation suppressed or enhanced? The present work would suggest that if the pop-in experiments are due to nucleation from surface defects that pop-in should be suppressed, not enhanced. This would merit deeper discussion.

Our Response:

We agree with the reviewer that there is no real understanding of the pop-in experiments. Our hypothesis is that there would be dislocation nucleation under the indenter tip as well as

interaction between hydrogen and pre-existing dislocations. The nucleation site is usually beneath the indenter tip in the bulk lattice [1], which is distinct from the surface nucleation in our study. So the hydrogen effect in the bulk sample with pre-existing dislocations can be very different from a nanostructure with large surface volume ratio.

We have added some discussion in our manuscript in Discussion part according to reviewer's suggestions.

[1] Schall, P., et al. "Visualizing dislocation nucleation by indenting colloidal crystals." *Nature* 440.7082 (2006): 319.

4. It is not clear that penta - twinned Ag nanowires are a representative material for studying hydrogen embrittlement. If I am not mistaken, the penta - twinned structure itself generates internal stresses. Thus there is some misfit stress distribution that will influence H absorption, dislocation nucleation, etc. The authors should discuss why these materials are a suitable model system, and address the issue of internal stresses.

Our Response:

It is not our intention to claim that the penta-twinned Ag nanowires are a representative material for studying hydrogen embrittlement in general, but it is indeed a good platform to study hydrogen embrittlement in dislocation starved systems including many types of nanostructures.

Out of all the synthesis methods of Ag nanowires, the method for synthesizing penta-twinned nanowires, the so-called polyol method, is most widely used as this method can produce large quantity of nanowires with high quality (e.g. uniform diameter and smooth surface) and well-controlled microstructure [1]. The electrodeposition method produces nanowires with relatively poor quality [2], while the physical vapor deposition method can only produce very small quantity [3], which presents a challenge for the experiment (i.e. hydrogen charging of the nanowires followed with picking up the single nanowires and placing them onto the testing stage). In addition, our previous work on penta-twinned Ag nanowires has shown that the deformation mechanism is governed by surface dislocation nucleation [4]. Other types of nanowires, such as those with grain boundaries, are expected to be studied in the future work.

The internal stress field may have some influence on hydrogen diffusion, but our SIMS results confirmed that hydrogen has diffused into the nanowires. The control group in this study consists of penta-twinned NWs with the same microstructure and hence internal stresses field, so it is suitable to study the effect of hydrogen in penta-twinned NWs.

[1] Zhu, Y., et al. "Size effects on elasticity, yielding, and fracture of silver nanowires: In situ experiments." *Physical review B* 85.4 (2012): 045443.

[2] Tian, M., et al. "Electrochemical growth of single-crystal metal nanowires via a two-dimensional nucleation and growth mechanism." *Nano Letters* 3.7 (2003): 919-923.

[3] Cheng, G., et al. "Anomalous Tensile Detwinning in Twinned Nanowires." *Physical review letters* 119.25 (2017): 256101.

[4] Qin, Q., et al. "Recoverable plasticity in penta-twinned metallic nanowires governed by dislocation nucleation and retraction." *Nature communications* 6 (2015): 5983.

5. The tensile tests were not performed within the hydrogen atmosphere. Thus, the actual hydrogen content would seem to be unknown - hydrogen generally desorbs from the metal quickly (this has plagued many bulk experiments, and should be even more problematic in NWs). Actual H may thus be limited to traps, which here would be the twin interfaces (although these are not usually traps; see recent results by Song and coworkers in Phys. Rev. Lett. on H trapping at grain boundaries) and to the sample surface. If I had to speculate, I would guess that there is very little H in the NWs, that the surfaces are mostly covered with H, and that the kinetics of H₂ formation on the Ag surface is controlling the actual sample conditions.

Our Response:

We must admit that it is a great challenge to identify the hydrogen desorption/effusion for an individual NW during the experiments. However, the SIMS measurements showed increasing hydrogen concentration in the NWs with increasing charging time. We agree with the reviewer that in our experiments, the NW surface was likely covered with H. In addition, some H must be diffused into the NWs, at least within certain depth into the NW surface. On the other hand, our experimental results after hydrogen charging showed clear effects of hydrogen, which indicates that hydrogen must be present in the NWs.

Since surface dislocation nucleation is the dominating deformation mechanism in our NWs, the hydrogen near the surface should play the critical role. We have run additional simulations where H atoms are only placed below the surface, consistent with the experiments. The simulation results agree well with the rest of simulations where hydrogen is distributed through the thickness as shown in Figure R2. In short, while the hydrogen concentration in the NWs is challenging to measure, the effect of hydrogen on dislocation nucleation – the main conclusion of the paper – should hold.

Figure R2 | MD simulation of tensile testing of penta-twinned NWs. Green line: hydrogen inserted close to the free surface can still delay surface dislocation nucleation.

6. In the simulations, the authors have introduced vacancies, which will serve as H traps and are also known to reduce the dislocation nucleation stress at surfaces. So, the simulations have been designed to create precisely nucleation sites that are then affected by H. So the results are then not too surprising (and see further comments below).

Our Response:

The presence of vacancies has been observed in our previous work and shown to play important roles in the time-dependent deformation of penta-twinned nanowires [1]. The effect of strain rate and temperature in SI Section S3 provides evidence for the importance of hydrogen diffusion in the simulation. For the nucleation sites, besides vacancy, surface notch (Supplementary Figure 5), and surface steps (Figure 4) were also discussed in the manuscript. In principle, our simulation results are consistent with a number of important previous studies on atomic mechanism in hydrogen embrittlement [2-4], and our work combining experiment and simulation provides the first study of hydrogen embrittlement in a dislocation starved system. While these findings may not appear surprising to some people, they are by no means trivial or straightforward. In fact, scientific discoveries often do not appear surprising once they are revealed and understood.

[1] Qin, Q., et al. "Recoverable plasticity in penta-twinned metallic nanowires governed by dislocation nucleation and retraction." *Nature communications* 6 (2015): 5983.

[2] Song, J., M. Soare, and Curtin, W. A. "Testing continuum concepts for hydrogen embrittlement in metals using atomistics." *Modelling and Simulation in Materials Science and Engineering* 18.4 (2010): 045003.

[3] Song, J., and Curtin, W. A. "Atomic mechanism and prediction of hydrogen embrittlement in iron." *Nature materials* 12.2 (2013): 145.

[4] Song, J., and Curtin, W. A. "Mechanisms of hydrogen-enhanced localized plasticity: an atomistic study using α -Fe as a model system." *Acta Materialia* 68 (2014): 61-69.

7. Figure 2 shows mainly a difference in deformation modes, with the stress - strain curves all being quite similar. The H leads to a very slight increase in nucleation stress, and only at 1.2% H. There is no information provided on the statistical scatter of these results. Since the nucleation is both thermally activated and influenced by vacancies, possibly vacancy clusters, and then H interactions with these defects, the trend shown by just three stress - strain curves is not convincing.

Our Response:

Thanks for the reviewer's suggestion. We did more simulations to exclude the possibility of thermal fluctuation in the increment of nucleation stress. For each hydrogen concentration, 5 different simulations were conducted. We have updated Figure 2b with averaged stress strain curves and error bars.

The new results with error bars clearly show the increment in the nucleation stress with the presence of hydrogen.

Figure 2 | MD simulation of tensile testing of penta-twinned Ag NWs. **a**, Simulated transition from distributed plasticity in the absence of hydrogen to localized plasticity in the presence of hydrogen. **b**, Averaged stress-strain curves of Ag NWs at different hydrogen concentrations with error bars.

8. The reason(s) for the localization of plasticity in the presence of H are unclear, and are not, from the data presented, necessarily systematic.

Our Response:

This is a very good question which has also been raised by the first reviewer. Only increment of the resistance to surface nucleation due to hydrogen is insufficient to induce localization of plasticity. There must be some non-uniform interactions between surface dislocation and hydrogen. The nonuniformity may exist in the initiation of surface nucleation stage, and/or during the succeeding nucleation stage.

The nonuniformity of hydrogen effect could contribute to the localization during the succeeding nucleation stage. In an elegant explanation of distributed plasticity in penta-twinned NWs [1], a dislocation nucleation event increases the local activation energy and immediately causes its nucleation site to be locked up, and further nucleation is shifted to a neighbouring site where activation energy has not been raised. In the presence of hydrogen, this situation could be different because hydrogen has caused a substantial increase in local activation energies. One can imagine that after a dislocation nucleation event, one source is activated and as soon as it is activated, the local activation energy drops because this site has been released from the constraint of hydrogen while all the other sites are still inhibited by hydrogen. This would then cause an avalanche effect in that the whole system fails at this weakened point. To support this hypothesis, the activation energy changes associated with succeeding dislocation nucleation were calculated based on the surface ledge model in Supplementary Material S-6. It was found that activation energy of local nucleation even decreases a bit in the presence of hydrogen, while activation energy of non-local nucleation increased significantly due to the effect of hydrogen. Overall, the presence of hydrogen substantially increases ΔE between non-local nucleation and local nucleation as shown in Supplementary Figure 9(b) and thus will clearly promote localized plasticity.

Furthermore, the presence of hydrogen promotes localized plasticity in another way that involves non-uniform hydrogen distribution. For the nonuniformity during the initiation of surface dislocation, we can assume that the activation energies at the surface nucleation sites follow a normal distribution during loading. The presence of hydrogen could, in addition to obviously increasing the mean value of the activation energies, increase the variance of the activation energies. The nonuniformity originates from the increment in the variance of the activation energies. Although the hydrogen is evenly distributed in the system initially, the diffusion and interaction between hydrogen atoms and nucleation sites are non-uniform. Supplementary Figure S10 shows the distribution of hydrogen on NW surface in MD simulation. While it does not show the activation energy distribution on the nucleation sites, non-uniform

distribution of hydrogen on NW surface can be observed. During initial nucleation, localization occurs if after the first dislocation nucleation, there is no nucleation at other sites within a δt time interval; in contrast, distributed plasticity occurs if after the first dislocation nucleation, within a δt time interval, there is nucleation at other sites. The probability of localization can thus be modeled as a nonhomogeneous Poisson process, with upper bound depending on the variance of the activation energy as

$$P_{localization} = (\max_{s'} \lambda(s', \tau_0)) \delta t \exp\{-\sum_s \lambda(s, \tau_0) \delta t\}$$

where s denotes a nucleation site and $\lambda = v_0 \exp\{-\frac{Q}{kT}\}$ the nucleation rate. Further details of the model are provided in Supplementary Material S-7. If λ is assumed to follow a log normal distribution, $\max_{s'} \lambda(s', \tau_0)$ becomes the critical term in (1) and $P_{localization}$ would increase with an enhanced variance in activation energy. MD simulations on a surface ledge model with different distribution of hydrogen were carried out in Supplementary Material S-8. This result clearly shows that at the initiation stage of the surface dislocation nucleation, the increment in the variance of the activation energies due to non-uniform interaction between hydrogen and the nucleation sites can lead to more localized nucleation.

Supplementary Figure 9 | NEB calculation of the nucleation activation energy change. (a) Two different nucleation configurations in the surface ledge model. (i), local nucleation: new partial nucleated at the extrinsic stacking fault created by the initial nucleation (blue dashed circle). (ii), non-local nucleation: new partial nucleated (blue dashed circle) at a surface ledge far away from the initial nucleation site where the extrinsic stacking fault was left. (b) Activation energy of two different scenarios with/without hydrogen as function of tensile strain.

Supplementary Figure 11 | Localization in the surface ledge model. (a) Hydrogen distribution in twelve surface ledges. The total number of hydrogen atoms are fixed, but the variance of concentration of hydrogen on surface ledges keeps increasing from (i)-(iv). (b) Snapshots of nucleation with different distributions of hydrogen from initial nucleation to 12ps after initial nucleation (extra 0.12% strain). The nucleation sites were marked by blue circles.

We have added this in the manuscript Discussion part.

New NEB calculation of activation energies of succeeding dislocation nucleation was added in SI Section S6.

Also detail of the statistical modeling is included in the SI Section S7.

New MD simulations on surface ledge model that reveal the transition was added in SI Section S8.

[1] Filleter, T., et al. "Nucleation-controlled distributed plasticity in penta-twinned silver nanowires." *Small* 8.19 (2012): 2986-2993.

9. As a reference, embrittlement in Ni is typically found at 0.1% (1000 ppm) (see recent work from Kumar et al., for instance; the authors might also examine very recent work from Somerday et al. on single vs. polycrystal Ni at comparable H content).

Our Response:

Thanks for pointing out these important experimental work in the literature. The hydrogen concentration is a very important parameter. In previous experiments in bulk system [1-3], the concentration of hydrogen is around 1000-3000 ppm. In our NWs sample, the SIMS results

showed that the hydrogen concentration is around 0-8000 ppm for different charging time. It is in the same order of the bulk material, while a little higher due to the large surface-to-volume ratio in NWs.

In the NW simulations, due to the time scale of atomic simulation, the hydrogen concentration we used is from 6000-10000 ppm, which is a bit higher than the experiment data. We need the hydrogen to diffuse to the dislocation nucleation sites within the simulation time scale, so the concentration of hydrogen is higher in this kind of dynamic simulation.

In the surface step simulation in Figure 4, the amount of hydrogen ranges from 1000 to 3000 ppm, which is consistent with the experiment data.

[1] Bechtle, Sabine, et al. "Grain-boundary engineering markedly reduces susceptibility to intergranular hydrogen embrittlement in metallic materials." *Acta Materialia* 57.14 (2009): 4148-4157.

[2] Lawrence, Samantha K., et al. "Effects of grain size and deformation temperature on hydrogen-enhanced vacancy formation in Ni alloys." *Acta Materialia* 128 (2017): 218-226.

[3] Tehranchi, Ali, and W. A. Curtin. "Atomistic study of hydrogen embrittlement of grain boundaries in nickel: II. Decohesion." *Modelling And Simulation In Materials Science And Engineering* 25.7 (2017): 075013. *Computational Materials* 3.1 (2017): 28.

10. I don't quite understand the importance of the stress recovery study. The experiments are nice, but I am not sure what is learned here. If there are dislocations/plasticity, there will be some recovery – this is already established in many nanomaterials (and bulk materials of course). If H suppresses nucleation, there will be fewer dislocations and hence less recovery. In the presence of H, diffusion to the dislocations will also pin them, leading to less recovery. The experimental observations are performed just below macroscopic yielding, so there would be few dislocations, thus limited dislocation network formation, and hence recovery even more likely. The simulations show these features. This part of the paper seems to mainly serve to demonstrate that H has some effects on dislocations. This has been demonstrated many times before, both experimentally and in simulations, and so the authors should identify what new insights are gained.

Our Response:

The recovery phenomenon indeed is less important in this study and is not the main purpose of Figure 3. More importantly, it is the stress relaxation during the *in-situ* holding that is found to be directly related to the surface dislocation nucleation [1]. In our current *in-situ* stress relaxation experiments, NWs with different hydrogen concentrations were relaxed at the same stress level, and we can find a clear drop in the magnitude of stress relaxation in the presence of hydrogen from Figure 3b. Since the stress relaxation is due to the sustained surface nucleation of dislocations, the attenuated stress relaxation further confirms that the presence of hydrogen has suppressed surface nucleation sources in the Ag NWs.

We have revised the Figure 3 by bolding the relaxation legend to remove the ambiguity and we have added the following sentence in the “***In situ* TEM stress relaxation testing**” part:

“The phenomenon of recoverable plasticity⁴⁵ was thus observed in penta-twinned NWs in the presence of hydrogen, and the change in the magnitude of stress relaxation at different hydrogen concentrations reveals an essential role of hydrogen on surface nucleation.”

[1] Qin, Q., et al. "Recoverable plasticity in penta-twinned metallic nanowires governed by dislocation nucleation and retraction." *Nature communications* 6 (2015): 5983.

11. Figure 4 shows that putting H atoms along the step at which nucleation occurs will cause nucleation to be suppressed. In this case, the nucleation moves to another step without H. This is not surprising. The same behavior was shown by Song et al. for H at a crack tip (an MSMSE paper, I I recall, that is not referenced here). Case d corresponds to nucleation not at the step, and so the process is shifted somewhat, giving rise to a different activation energy vs. normalized stress in part f. This is an indirect effect of H, not a direct effect. Nucleation away from the stress concentration, when it is blocked sufficient by H, was shown by Song et al. in the same paper. Results shown in SI Fig. 7 are also related to results shown in the same paper.

Our Response:

The simulation setup was indeed inspired by the previous breakthrough works in atomic mechanism in hydrogen embrittlement by Song and Curtin [1-3], and it is not surprising that our conclusion is consistent with the atomic mechanism proposed by the authors. But there are some differences. Specifically, in our NW system, the existing defects for surface nucleation were not crack tip but surface steps as observed in our TEM study. The effect of hydrogen on dislocation nucleation was directly studied on a simplified surface step system to better support our experiment observation.

We have included the important references that were missed before [2-3].

[1] Song, J., and Curtin. W. A. "Atomic mechanism and prediction of hydrogen embrittlement in iron." *Nature materials* 12.2 (2013): 145.

[2] Song, J., and Curtin. W. A. "Mechanisms of hydrogen-enhanced localized plasticity: an atomistic study using α -Fe as a model system." *Acta Materialia* 68 (2014): 61-69.

[3] Song, J., M. Soare, and Curtin. W. A. "Testing continuum concepts for hydrogen embrittlement in metals using atomistics." *Modelling and Simulation in Materials Science and Engineering* 18.4 (2010): 045003.

12. The supplementary simulations at different T show that H can soften the material at T=100K (for the one simulation shown). There is no interpretation of this result, which would seem to be important. That is, H must have two effects, one that occurs without diffusion and softens the material and the second with diffusion that strengthens the material. The softening would seem to align better with the nanoindentation pop - in results.

Our Response:

We have discussed the temperature and strain rate effect in SI-S3. If the simulation strain rate is high (i.e., 1×10^8 /s) or the temperature is low (see SI Section S3, Supplementary Figures 4 and 5), the diffusion of hydrogen will be limited, thus its interactions with potential nucleation sites will be limited. Hydrogen atoms will act as sessile inclusions and the lattice distortion will reduce the threshold for dislocation nucleation, which is consistent with the conclusion of the indentation simulation [1].

As the reviewer pointed out, hydrogen must have two effects, which depend on the diffusion time scale of hydrogen. This is one key point we learned from the simulation results and emphasized in our manuscript. It is also the reason we adopted extremely low strain rate and high temperature in our tensile simulation.

We also agree with the reviewer that the softening aligns better with the nanoindentation pop-in results, but as described in the reference [1], this lattice swelling induced softening is only to a limited extent, and a full understanding of the pop-in results still needs further study. We also want to mention that the strengthening of hydrogen in the surface nucleation is not in conflict with the softening effect in homogeneous nucleation in the bulk.

We have further revised SI Section S3 and Supplementary Figures 4 and 5 to better discuss the temperature and strain rate in the effect of hydrogen.

[1] Zhou, X., et al. "Atomistic investigation of the influence of hydrogen on dislocation nucleation during nanoindentation in Ni and Pd." *Acta Materialia* 116 (2016): 364-369.

13. I could envision that these MD nanowires have significant surface and internal stresses, due to the small size (even if the misfit due to the pentatwin structure were to be absent or small). The introduction of H could change these internal and surface stresses (mechanics only), changing the required applied stress for nucleation. This would also make the results diameter dependent. This has been observed in a number of nanowire - type simulations. H diffusion/segregation would then further be driven by these stress differences. This requires further discussion, since it pertains to interpreting experiments as well.

Our Response:

We agree with the reviewer that the nucleation stress in nanowires is diameter dependent, as reported in many simulations and experiments; some experiments specifically on penta-twinned Ag nanowires can be found [1-2]. However, the distributed plasticity was found to be dominating in all penta-twinned NWs with diameter smaller than 100nm [2], the diameter only has quantitative changes and does not affect the mechanism we found in this manuscript. Therefore, it is not our intention to investigate the size effect in this context. The surface stress may have effect on the diffusion property of hydrogen, but this also doesn't qualitatively change the interaction between hydrogen and surface dislocation. In the present work, we intentionally used the one size of nanowires to avoid size effect. The comprehensive size-dependent behavior will be studied in the future.

[1] Zhu, Y., et al. "Size effects on elasticity, yielding, and fracture of silver nanowires: In situ experiments." *Physical review B* 85.4 (2012): 045443.

[2] Filleter, Tobin, et al. "Nucleation-controlled distributed plasticity in penta-twinned silver nanowires." *Small* 8.19 (2012): 2986-2993.

14. There were insufficient details provided about these simulations. Is the potential here reasonably representative of the Ag - H systems? What is the migration barrier for H atoms in this Ag - H system?

Our Response:

About the atomic potential, the structure, elastic properties and defect energies for silver were fitted to experimental and *ab initio* measurements. Binding and migration energies of hydrogen were fitted to the experiment and *ab initio* measurements of Pd-Ag-H alloys. The corresponding potential data and reference properties can be found at the website of the NIST Interatomic Potentials Repository Project.

We have calculated the migration energy barrier of hydrogen between the nearest octahedral and tetrahedral sites through the NEB method. It shows that tetrahedral to octahedral has the lowest barrier for migration (0.0154 eV). The energy barriers of other migration paths are: from octahedral to tetrahedral: 0.55 eV, from tetrahedral to tetrahedral: 0.098 eV and from octahedral to octahedral: 0.62 eV.

The migration energy barrier in this system is close to the experimental measurements [1-2].

[1] Ishikawa, T., and R. B. McLellan. "The diffusivity of hydrogen in the noble metals at low temperature." *Acta Metallurgica* 33.11 (1985): 1979-1985.

[2] Kurokawa, Hitoshi, et al. "Development of new kinetic Monte Carlo simulator for hydrogen diffusion process in palladium-silver alloys." *Applied surface science* 244.1-4 (2005): 636-639.

15. Does H have any attraction to the twin boundaries? What are the binding energies of H to the vacancies and/or steps, and is this realistic? 1% vacancies is extremely high as a practical matter, which influences the pure Ag results as well as Ag - H results, but this is not discussed. I don't have a problem with using 1% for test purposes, but without detailed examination, it is unclear what effects are determining the MD results. I presume the NWs were created with perfectly flat facets, aside from the vacancies, but then Figure 4 studies explicit steps. So, if the authors argue nucleation is at steps (perfectly acceptable – see nanoscale experiments in Au by Weinberger et al (NatComm 2010)), then what is the relevance of the results in Figure 2? As with other aspects of the paper, the short format here does not enable sufficient discussion of the results.

Our Response:

No, according to our previous simulation, the twin boundary is not a preferred site for hydrogen. In contrast, in the current system, the binding energy of hydrogen to vacancy is: 2.46 eV and to

step is: 1.785 eV. This implies that vacancies/steps can strongly trap H in our system, which is consistent with previous DFT and experiment conclusion. Although the EAM potential overestimates the binding energy when quantitatively comparing with DFT data in the reference but they are still in a realistic range.

Due to the time scale of atomic simulation, the vacancy was introduced as pre-existing defects that will facilitate dislocation nucleation. One of our conclusions is that the effect of hydrogen cannot be achieved in perfect crystalline structure in atomic simulations. To demonstrate the effect of hydrogen, artificial defects were introduced here. Surface step or notches can also be introduced to the NW surface and are expected to show similar results, as showed in Figure 3c and 4. However, in the full size tensile simulation, introducing surface steps or notches will bring much more complexity in the problem, such as number of steps, density of steps and arrangement of the steps. To simplify the problem, uniformed vacancy defects were introduced in the full-size tensile NW model in Figure 2 to serve as initial defects that trigger surface dislocation nucleation. The tensile simulation results are consistent with our *in-situ* experiment observations.

Interactions between hydrogen with surface steps were carefully modeled. Increment in the activation energy was quantitatively calculated in the simplified quasi-2D model, as shown in Figure 4.

Reviewer #3 (Remarks to the Author):

Dear authors,

You have done good experimental work and combine it perfectly with the simulation to achieve very good results. I enjoyed the reading as it gives new interesting insight on the mechanism of hydrogen embrittlement. Unfortunately, there are some minor mistakes which need in some work. See the following comments.

Response:

We would like to thank the reviewer for

1. Comment: Figure 1a: Give scale bar length in drawing and not only in the figure caption

Our Response:

Thanks for the suggestion; however, it is suggested to list the scale bar length in the figure caption according to the journal's style.

2. Comment: Figure 1b: Give the strain rate of the tests also in figure caption.

Our Response:

The loading strain rate for *in situ* TEM tensile testing was $\sim 0.005\%$ /s. We have added this in the figure caption.

3. Comment: Figure 1c and 1d: Explain the numbering “i” to “vi” in the figure caption or directly one the picture. You explain it in the text from line 96 to line 101 but it is not understandable, when looking solely on the figure. Furthermore, in the text you explain that “I” to “iv” represent different load levels (or strains). Could you also give the absolute values of load or strain which corresponds to single pictures i to iv.

Our Response:

We have added the corresponding strain for each snapshot (i-iv) in the figure caption.

4. Comment: Figure 1c and 1d: Give the size of the scale bar in the figure.

Our Response:

It is suggested to list the scale bar length in the figure caption according to the journal's style.

5. Comment: Supplementary figures and tabular. I received the supplementary information and the paper in two different files. I suggest to combine those and to add the supplementary information as appendix in the main paper file for publishing. This helps the reader not to

oversee any information when downloading the file once published, especially as you give very helpful information in the supplement to answer arising questions, e.g. the hydrogen content. I am aware that you have to check with the politics of the magazine.

Our Response:

The journal has a length limit for an article so that some of the results are put into the supporting information. The SI file can be easily downloaded from the website if the manuscript is accepted to publish.

6. Comment: Hydrogen concentration after charging: The normalized hydrogen concentration is given with the uncharged sample as reference value. Could the absolute (ppm) hydrogen concentration be given?

Our Response:

We listed the absolute values (wt ppm) here. The as-received sample was examined with a low value of hydrogen. For the comparison, we used the relative value to emphasize the change of hydrogen under different charging time.

Charging time	c_H	
	at.%	wt ppm
hrs		
0	0	0
12	0.15	14
24	0.41	38
36	0.63	59
48	0.8	75

Supplementary Table 1 | Hydrogen difference (c_H) in Ag NWs with different charging time into a H₂/Ar (molar ratio, 1:1) atmosphere, normalized to the as-received one (not immersed in H₂/Ar atmosphere).

7. Comment: Concerns Fig 2a: Can you give a scale bar? What are the dark and green areas? Am I right that dark areas are distributed plasticity and green areas are without plasticity.

Our Response:

Sorry for the confusion. The penta-twinned NW samples were 15 nm in diameter and 80 nm in length. We have added a scale bar in Figure 2a.

Figure 2a shows atomistic simulation results, where atoms were colored according to their crystalline structure – green for face-centered cubic symmetry, red for hexagonal close-packed (HCP) symmetry and gray for the atoms at dislocation cores, surfaces and point defects. This information was in the Method part.

In the figure with only HCP atoms shown, the dark red flecks are stacking faults left in the NW after surface dislocation nucleation. In the left part of Figure 2a without hydrogen, dislocation nucleation distributed along the whole NW, while in the right part of Figure 2a with hydrogen, dislocation nucleation was localized.

8. Comment: Fig 2b: Is it possible to use different lines styles, e.g. full line or dotted line, to allow a color-blind reader to distinguish between the lines.

Our Response:

Thanks for the kind suggestion, we have revised Figure 2b accordingly.

9. Comment: Fig 3c: I took me some time to realize that the dark areas on the penta-twinned NWs represent dislocations. Please add this information in the figure caption. Furthermore, the legend gives the hydrogen concentration with no hydrogen on top and maximum hydrogen at the bottom. It would be more logical for me to sort the legend in the order of the stress-relaxation-lines in the figure.

Our Response:

Thanks for the suggestion. We have added more detail information in the figure caption and changed the order of legend.

10. Comment: Fig 4e: Similar to the last comment. I would sort the legend according to the lines. Case d on top as it is the curve on top.

Our Response: Thanks for the suggestion. We have changed the legend order accordingly.

11. Comment: Line 91: You say that the NWs vary in diameter by about $\pm 4\%$. Is there any effect of this variation of the cross section on the elastic-plastic behavior the NWs? You may add a small comment on this in the paper.

Our Response:

The mechanical properties are size-dependent as reported in previous works [1-2]. Here we chose the NWs with nearly identical diameters to avoid the size effect on the mechanical behavior.

We revised in our manuscript as:

“It has been reported that mechanical properties of penta-twinned Ag nanowires are size dependent [refs]. All of the tested NWs in this work had similar diameters of 71 ± 3 nm to focus the study on the hydrogen effect without the size effect”

[1] Zhu, Y., et al. "Size effects on elasticity, yielding, and fracture of silver nanowires: In situ experiments." *Physical review B* 85.4 (2012): 045443.

[2] Filleter, Tobin, et al. "Nucleation-controlled distributed plasticity in penta-twinned silver nanowires." *Small* 8.19 (2012): 2986-2993.

12. Comment: Line 91-92: You refer to the "Methods". Help the reader to find the "methods"-chapter. You may say that there is an appendix.

Our Response:

The journal asks for a strict format for publication. So we use "Methods" here.

13. Comment: Line 99: You write that necking is observed in the NWs. Can you give the reduction of area?

Our Response:

The necking was clearly observed in the as-received samples. From the TEM characterization, the maximum reduction of the cross-sectional area (Figure R3) was ~15%.

No obvious necking was captured for the NW with long charging time (>24 hrs) before the failure of the NW.

Figure R3. Postmortem TEM characterization of the as-received Ag NW after the tensile testing. **a** is a magnified TEM image showing the obvious necking in the NW, corresponding to the selected area in **b**. The multiple neckings were marked by green dashed ovals.

14. Comment: Line 121: You write "Figure 3 shows snapshots of the observed failure modes". Are you sure you mean figure 3 and not 1b. If you mean figure 3 you should explain in the text what to see in figure 3 when referring to it as it is used here first time in the text.

Our Response:

Sorry for the typo here. It should be Figure 2a. We have changed this in the manuscript.

15. Comment: You give the hydrogen concentration only in the supplements. I think this information is important to rate the results. I suggest adding this information to the main part of the paper.

Our Response:

We have listed the hydrogen concentrations in the right bottom of Figure 1b corresponding to the different charging time.

16. Comment: Line 125m, Chapter about the tensile test. Are you sure you want to refer to Figure 3b? For me it is more logical to see the higher yield stress in the stress-strain curves in Figure 1b?

Our Response:

Sorry for the typo here. It should be Figure 2b. We have changed this in the manuscript.

17. Comment: Concerns the tensile test. You do not give the strain rate in the main part of the paper but only in the methods chapter. For bulk material tests the strain rate can – especially when there is HE- have a big effect on the fracture strain. Mostly the lower the strain rate, the higher the HE and the lower the strain at fracture. Compare standard ASTM G 129 which considers this behavior in an international standard. Also the chosen strain rate of 5×10^{-5} 1/s is, at least for some bulk material, a rather high strain rate, when looking on the effect of HE in mechanical test. Could you explain why you have chosen this strain rate? Did you do test with other strain rates to see the strain rate effect?

Our Response:

In experiments, we did the test in a quasi-static loading with a fixed strain rate ($\sim 0.005\%$ /s) for all the tested NWs. We did choose the fixed strain rate for the in situ TEM tensile testing so as to observe the microstructure evolution simultaneously. We will do further study on the strain rate effect in the future work.

18. Comment: Concerns the hydrogen desorption. This comment might be linked to the answer of the last comment concerning the strain rate. I understand that you have done the tensile tests and the relaxation test on hydrogen pre-charged specimen. Normally, at least in bulk steel, after hydrogen pre-charging, the hydrogen desorption/effusion starts directly after the end of charging. Is there hydrogen effusion in gold NWs? When there is effusion, what is the effusion rate? How much hydrogen desorbs between the end of charging and the start of the test. I suspect that the tests in the TEM are done in vacuum and this should support the hydrogen effusion. How much hydrogen remains in the NWs after the end of the mechanical tests? I expect some proof that there is hydrogen in the NWs when testing

Our Response:

It is a great challenge to identify the hydrogen desorption/diffusion for an individual NW. Thus, we examined the hydrogen concentration of the samples with a number of NWs immediately after immersing the NWs into an Ar+H₂ atmosphere. We did observe the difference of hydrogen concentration of the sample with different charging time. And such difference will cause the change of the mechanical properties of the NWs with different charging time as shown in Figure

1b. A high concentration of hydrogen left in the NW will cause the increase of tensile strength and the decrease of the plastic strain.

19. Comment: Line 148 and Fig. 3a: Can you give the strain rate and the hold time at maximum and minimum load. In the text (line 148) you name the test “tensile test” and in the figure caption (Fig. 3a) “relaxation test”. Those are relaxation tests (or hysteresis loops). Use the same name throughout the paper.

Our Response:

In experiments, we did load or unload with a fixed strain rate ($\sim 0.005\%$ /s) for all the tested NWs. Both the hold time (at maximum load) and relaxation time (at minimum load) were 15 minutes.

We did stress relaxation during the tensile test by holding the NW at a certain strain. Thanks for the suggestion. We have changed the “tensile test” to “relaxation test” in this part for consistency.

20. Comment: Chapter MD simulations of dislocation nucleation and activation energy, starting line 165: You give simulation evidence and experimental results that “hydrogen can suppress surface dislocation nucleation (authors cited in line 166-167)” in metallic nano wires.

I miss direct experimental evidence in the paper. I expect some TEM pictures of dislocations structures to proof this. Would it be possible to make TEM lamellas with FIB from the tested NWs with different hydrogen concentration to analyze the dislocation structures? You could compare the dislocations structures of the different NWs in the area of necking and outside the necking area. Then you would have direct evidence that the hydrogen reduces the dislocation nucleation. Alternatively, you could cite references in which this mechanism is shown.

Our Response:

Our experimental result of stress relaxation in Figure 3b can be seen as the evidence to support our conclusion. In our previous work [1], we have observed directly by in-situ TEM and atomistic simulations that this stress relaxation was induced by surface dislocation nucleation. In the present study as shown in Figure 3b, the stress relaxation became attenuated as the hydrogen concentration increased, which suggests that the presence of hydrogen has suppressed surface nucleation sources in the Ag nanowires. This evidence, although not as directly as TEM image of dislocation structures, can support our hypothesis. It is very challenging to cut a piece of sample from the tested NW owing to the limited sample size.

[1] Qin et al. "Recoverable plasticity in penta-twinned metallic nanowires governed by dislocation nucleation and retraction." *Nature communications* 6 (2015): 5983.

Reviewers' comments:

Reviewer #1 (Remarks to the Author):

The authors have addressed most of my review comments. I am happy to recommend the publication of this paper.

Reviewer #2 (Remarks to the Author):

The authors have discussed all of the issues raised in the first review, but little of it is incorporated into the main text and the critiques in the first review generally remain unanswered in the paper.

The main point of the paper is to show that nucleation of dislocations can be affected by Hydrogen. This point is not new, neither experimentally (e.g. pop-in) nor via simulation (e.g. crack tip). The specific demonstration here is also in a very specialized nanomaterial and so is not generally translated to a broader context.

Neither experiments nor simulations are quantifiable. The authors do not know the H content of their real materials, and they acknowledge that the effects are probably due to surface H. The simulations are not done under realistic conditions, and the main features are in any case driven by H segregated to artificial (vacancies) or intrinsic (surface step) defects, leading to changes in nucleation.

The work is therefore not definitive and has little to do with H embrittlement in other materials. The paper is - as stated in the first review - certainly a contribution to the evolving experimental and simulation literature on effects of H on various plasticity phenomena. As such, it merits publication in journals where similar work has appeared frequently (such as Acta Materialia) since it is at the same level. By publishing a short format paper, where all the caveats and uncertainties are buried in Supplementary Material (rather than merely provided more details of the work reported in the main text), the paper can give the impression that something more significant has been accomplished relative to the literature.

If Nature Communications is intended to highlight fairly definitive results of broader interest and/or new phenomena, then this paper would not satisfy those requirements, in my opinion.

Reviewer #3 (Remarks to the Author):

Dear authors,

well done. The review process enhanced the paper. It is easier to understand, the reader can follow your work and he is enabled to redo it. Therefore, I suggest to the journal to publish the paper. I have two minor remarks. You added a scale bar in Figure 2a as I suggested. Please also give the size of the scale bar. At least I did not find it in the pdf-file. Secondly in line 102 you write "in the tested NWs. Figure 1c shows that for" In the pdf it looks as there is a double space-bar after the point of sentence, before "Figure 1c".

Best regards

Response to the Reviewer Comments:

Reviewer #1:

The authors have addressed most of my review comments. I am happy to recommend the publication of this paper.

Response: Thank you for recommending publication of this paper.

Reviewer #2:

The authors have discussed all of the issues raised in the first review, but little of it is incorporated into the main text and the critiques in the first review generally remain unanswered in the paper.

Response: We are happy that the reviewer found that we have discussed all of the issues raised in the first review. It is not true that “little of it is incorporated into the main text”. A number of key issues along with substantial details have been incorporated into the main text. Of course, some additional details have been incorporated into the Supporting Information that is still an integral part of the paper. For papers published in Nature Communications, it is not uncommon to place some of the details in the Supporting Information, which is an integral part of the journal. We do not agree that the critiques in the first review generally remain unanswered in the paper, which appears to be contradictory to what the reviewer just said. Moreover, in case some of the critiques indeed remain unanswered, please point them out and we will be happy to address.

The main point of the paper is to show that nucleation of dislocations can be affected by Hydrogen. This point is not new, neither experimentally (e.g. pop-in) nor via simulation (e.g. crack tip).

Response: We respectfully disagree with the reviewer. For the first time we have devised a clean system to probe H embrittlement. The H effect on dislocation nucleation, only predicted in simulations in the case with crack tips by Curtin and co-workers (which is not exactly the same as a nanowire), has not been observed in experiments before. The pop-in experiment was different. The pop-in experiment was conducted on bulk materials using nanoindentation. Contrast to the strain hardening found in our case, softening in the pop-in stress during nanoindentation was observed in the presence of hydrogen. To the best of our knowledge, this observation has not yet been clearly understood.

Apparently, the reviewer’s comments on pop-in experiments are in contradiction to his/her own comment in the first round. By the way, we have addressed the reviewer’s original comments, which we assume are satisfactory to the reviewer. Just quote the comments here – “It is unclear whether nanoindentation pop-in experiments are related to actual nucleation. The referenced paper 26 showed that the referenced paper 25 incorrectly interpreted simulations. The referenced paper 29 presents a model where there is a pre-existing loop, around which H atoms can aggregate. This is thus not nucleation as usually envisioned, nor as studied here. Currently there

is no real understanding of the pop-in experiments.” “Note that the pop-in experiments, if real nucleation, expressly contradict the current experiments. So, is nucleation suppressed or enhanced? The present work would suggest that if the pop-in experiments are due to nucleation from surface defects that pop-in should be suppressed, not enhanced. This would merit deeper discussion.”

Our findings, combining experimental observations, atomic simulations and theoretical modeling, all revealed a totally new and distinct phenomenon of H effect at the nanoscale with the underlying mechanism clearly explained. We would like to emphasize that different from the pure simulation study of crack tip nucleation, our state-of-the-art in-situ experiments clearly revealed the H effect on surface nucleation as manifested by the transition from distributed plasticity to localized necking.

The specific demonstration here is also in a very specialized nanomaterial and so is not generally translated to a broader context.

Response: In nanomaterials, surface plays a key role in dislocation activities, which bears a close similarity to bulk materials with free surfaces (e.g. cracks) or interfaces. In this sense, our results can be generally extended to a broader context. Furthermore, our demonstration is not in a very specialized nanomaterial. While we focused on a specific type of nanomaterial, penta-twinned silver nanowires, to illustrate our finding on the H effect, we did perform additional work on another type of nanomaterial, single-crystalline silver nanowires. We found the same effect – H suppresses dislocation nucleation and as a result increases the yield strength. Note that it is experimentally very challenging to synthesize nanowires with high quality and well-defined microstructures. Currently we only have access to these two types of nanomaterials (with the best quality to our knowledge), but we believe the H effect should hold, at least, for most other FCC nanomaterials. Indeed our simulations have confirmed that.

We have added Figure R1 in the supplementary materials.

Figure R1. Stress-strain responses (a,b) for single crystalline Ag NWs at different H concentrations. b shows the details of stress-strain curves at a magnified strain scale from 0 to 0.04 as shown in a. The strain rate for in situ TEM tensile testing was $\sim 0.005\%/s$. c, Cross-sectional TEM image of a single crystalline Ag NW and the corresponding diffraction pattern taken from $\langle 110 \rangle$ zone axis. Scale bar, 20 nm.

Neither experiments nor simulations are quantifiable. The authors do not know the H content of their real materials, and they acknowledge that the effects are probably due to surface H. The simulations are not done under realistic conditions, and the main features are in any case driven by H segregated to artificial (vacancies) or intrinsic (surface step) defects, leading to changes in nucleation.

Response: In our NWs sample, the SIMS results showed that the hydrogen concentration is around 0-8000 ppm for different charging time, which is in the same order as that of the bulk material. In the simulations, due to the time scale of atomic simulation, the hydrogen concentration we used is from 6000-10000 ppm, which is a bit higher than the experiment data. This is because we need the hydrogen to diffuse to the dislocation nucleation sites within the simulation time scale, so the concentration of hydrogen is higher in the simulation.

In addition to H adsorbed on nanowire surface, H rapidly diffuse into the nanowire due to the small nanowire diameter. Our simulations with H atoms only placed below the surface, consistent with the experiments, showed strong suppression effect on dislocation nucleation.

The work is therefore not definitive and has little to do with H embrittlement in other materials. The paper is - as stated in the first review - certainly a contribution to the evolving experimental and simulation literature on effects of H on various plasticity phenomena. As such, it merits publication in journals where similar work has appeared frequently (such as Acta Materialia) since it is at the same level. By publishing a short format paper, where all the caveats and uncertainties are buried in Supplementary Material (rather than merely provided more details of the work reported in the main text), the paper can give the impression that something more significant has been accomplished relative to the literature.

Response: As detailed in the point-to-point response above, we believe our work is as definitive as it can be based on the state-of-the-art experimental and modeling capabilities, and our findings make a novel and important contribution to the field. The reviewer's overly harsh criticism is not based on facts and in fact not shared by the other two reviewers (and even in a few cases contradictory to his/her own comments in the first review). As quoted from Reviewer #2, "H embrittlement is a complicated problem that has resisted clear understanding for a long time," we believe the new approach that we have taken in this work, using nanomaterials instead of complicated bulk materials that have been studied for many decades, will provide new insights to the long-standing problem. In particular, because of known microstructure in our samples, our results are definitive as supported by quantitative in-situ TEM results, atomistic simulations and theoretical modeling. Following this bottom-up approach, we can introduce more complicated microstructures step by step (e.g. from single crystalline nanowires to polycrystalline ones). We hope this will provide progressive but concrete understanding of H embrittlement.

Reviewer #3:

Dear authors,

well done. The review process enhanced the paper. It is easier to understand, the reader can follow your work and he is enabled to redo it. Therefore, I suggest to the journal to publish the paper. I have two minor remarks. You added a scale bar in Figure 2a as I suggested. Please also give the size of the scale bar. At least I did not find it in the pdf-file. Secondly in line 102 you write “in the tested NWs. Figure 1c shows that for” In the pdf it looks as there is a double space-bar after the point of sentence, before “Figure 1c”.

Response: Thank you for your kind words and finding that the review process enhanced the paper. We are grateful to all the reviewers as their comments indeed enhanced the paper.

Regarding the two comments, the scale bar is 7 nm that we have added in the revision. We have removed the double space-bar in line 102 as pointed out.

Reviewers' comments:

Reviewer #4 (Remarks to the Author):

This manuscript presents a combined experimental/modeling investigation of mechanical deformation in pentatwinned nanowires (NWs) after charging in a H atmosphere. The authors conclude that hydrogen makes surface nucleation of dislocations more difficult, which eventually leads to more rapid flow localization and failure. In addition to reviewing this paper, I have been asked to offer an opinion on the previous three reviews.

The work in this manuscript has been carried out well. The results are novel and mostly reliable. However, as I will argue below, one part of it is flat wrong, namely the values of hydrogen concentration reported in the NWs. Unfortunately, this piece of information is a keystone element in the manuscript's narrative: pull it out, and the whole story collapses. Without it, there is no basis to claim that the experimentally-determined changes in mechanical properties are due to hydrogen embrittlement. Without it, the rationale for the modeling is removed and the relevance of the model to the experiment vanishes.

So the flaw is rather severe and I certainly do not think that the manuscript can be published as it stands. On the other hand, it is only severe to the overall interpretation and narrative as presented here, not to the mechanical testing or atomistic modeling results themselves. Thus, I think that the authors should be able to salvage most of the work that has gone into this study, perhaps by coming up with a different interpretation or by publishing the experimental and modeling work separately. Such changes would probably require withdrawal of the manuscript, a thorough rework, and resubmission (to NC or elsewhere) in the form of a new manuscript (or manuscripts).

Now the reasons why the reported H concentration is wrong:

- Thermodynamics. Sieverts law allows us to compute the equilibrium H concentration under the charging conditions used in this study (NB: such a calculation should have been performed and reported). The authors charge in 50/50 Ar/H₂ atmosphere. Although they do not say so explicitly, I take it the atmosphere was at room temperature and pressure. The formation energy of H interstitials in Ag is not reported by the authors. However, I know it to be larger than that of H in Ni, which is 0.17eV. Thus, I'll use the Ni value to come up with an upper bound estimate for the equilibrium H concentration. Following the expressions used by Bechtle et al. (Acta Mater. 57, 4148 [2009]), I arrive at an expected H concentration of 4 appm (~0.04 wppm). This value is orders of magnitude smaller than the ones reported by the authors (up to 75 wppm, i.e. ~8000 appm), which are high even when compared to those achievable under high pressure/high temperature H charging. For example, to achieve a H concentration of 3400 appm in Ni, Bechtle et al. had to charge in a pure H₂ atmosphere at 138 MPa and 200 C. In short, the H concentrations reported here are unrealistic for the charging conditions used.
- Kinetics. In their response to one of the reviewers, the authors report a H diffusivity in Ag of $1e-14 \text{ m}^2/\text{s}$. The diameter of the Ag NWs is about 20nm. Thus, using d^2/D , we find that the characteristic diffusion time of H into and out of the NWs is less than 1s (0.04s, for what it's worth). Thus, for all practical experimental purposes, the H concentration in the NWs comes to equilibrium with the surrounding atmosphere instantaneously. By contrast, the variations in measured H concentration reported in supplementary figure 1 occur over times ranging up to 48 hours. Thus, these variations cannot be ascribed to the charging of the NWs. Furthermore, with a characteristic outgassing time of less than 1s, it is safe to say that any H that may have been charged into the NWs is completely gone by the time the mechanical tests are performed.
- Measurement method. The authors repeatedly state (in the manuscript and in their responses to the reviewers) that the best method of measuring H concentrations is SIMS. However, it is important to realize that no flavor for SIMS (including in situ TOF-SIMS) has an areal footprint smaller than about a square micron. Indeed, most have footprints that are much larger than that. By contrast, a vertically aligned, 20nm-diameter NW presents a cross-section area of three ten-thousandths of a square micron! The signal returned by SIMS from a measurement performed

upon such a NW will be utterly dominated by the surrounding background (presumably, the substrate upon which the NW was grown). Most NWs are grown as forests, so perhaps a square micron-sized SIMS measurement will encompass several NWs, but that will still not lead to a meaningful NW signal in the SIMS measurement unless the spacing between the NW is comparable to (ideally: smaller than) the NW diameter. The authors do not report the spacing between NWs in their samples, but based on reports by others, I expect that it is much larger than the NW diameter. Thus, SIMS is not suitable for measuring H concentration in NWs. In the present study, it most likely measures H uptake into the substrate.

In light of the foregoing, I do not think that the narrative about hydrogen embrittlement presented in this manuscript is credible. On the other hand, the mechanical tests (which are impressive in their own right) clearly do show that something changes in the mechanical response of the NWs. This is an interesting finding, but I think more work is needed to determine the reasons for these changes. Similarly, the modeling work is top notch. However, for the reasons presented above, I do not believe it is relevant to the mechanical tests.

This manuscript underwent thorough review by three other referees. All three raised the same concern about the H concentration as I did. However, unlike the previous reviewers, I do not find the authors' responses to these criticisms to be satisfactory.

However, for what it's worth, I think the authors' responses to all the other concerns are entirely adequate. In particular, I do not share the second reviewer's view that the work presented by the authors lacks novelty.

Response to the Reviewer Comments:

Reviewer #4 (Remarks to the Author):

This manuscript presents a combined experimental/modeling investigation of mechanical deformation in pentatwinned nanowires (NWs) after charging in a H atmosphere. The authors conclude that hydrogen makes surface nucleation of dislocations more difficult, which eventually leads to more rapid flow localization and failure. In addition to reviewing this paper, I have been asked to offer an opinion on the previous three reviews.

The work in this manuscript has been carried out well. The results are novel and mostly reliable. However, as I will argue below, one part of it is flat wrong, namely the values of hydrogen concentration reported in the NWs. Unfortunately, this piece of information is a keystone element in the manuscript's narrative: pull it out, and the whole story collapses. Without it, there is no basis to claim that the experimentally-determined changes in mechanical properties are due to hydrogen embrittlement. Without it, the rationale for the modeling is removed and the relevance of the model to the experiment vanishes.

So the flaw is rather severe and I certainly do not think that the manuscript can be published as it stands. On the other hand, it is only severe to the overall interpretation and narrative as presented here, not to the mechanical testing or atomistic modeling results themselves. Thus, I think that the authors should be able to salvage most of the work that has gone into this study, perhaps by coming up with a different interpretation or by publishing the experimental and modeling work separately. Such changes would probably require withdrawal of the manuscript, a thorough rework, and resubmission (to NC or elsewhere) in the form of a new manuscript (or manuscripts).

Response: Thanks for the critical question about the high values of hydrogen concentration in the charged NW samples. After careful examination of our experimental conditions and a thorough study of the literature, we concluded that the high values obtained from the SIMS experiments are attributed to the surface adsorption of Ag NWs. When preparing the SIMS samples, we dispersed a thick layer (thickness of ~0.5 mm) of Ag NWs on a Si substrate, which allowed detection of Ag and H from the NWs (not from the substrate) in the SIMS experiments. The Ag NWs are randomly oriented forming a dense yet hydrogen permeable network (see Figure R1). For more information about the Ag NW network, please refer to Ref. [R1]. Before hydrogen charging, the substrate containing Ag NWs was purged for 2 hrs in a vacuum chamber with a constant flow (~0.1 sccm/hr) of pure Ar (99.99999%) at 250 °C, to remove the surface contaminations such as hydrocarbon. The treated samples before hydrogen charging were examined in SIMS showing almost no hydrogen. As pointed out by the reviewer, a high concentration of hydrogen could not be left in the bulk lattice of Ag according to the high diffusivity of hydrogen in Ag ($4.9 \times 10^{-12} \text{ m}^2/\text{s}$) at room temperature and high formation energy of H interstitials. But in our experiments, the NW networks showed a very high surface/volume ratio compared with the bulk counterpart. The high values of hydrogen concentration in the charged samples were most likely due to surface adsorption during the charging process. We will provide a detailed explanation about the surface adsorption in the following responses. And we have revised the atomic simulations accordingly.

Figure R1. A typical SIMS sample that is a network with randomly oriented Ag NWs.

[R1] Xu, F. and Zhu, Y., "Highly conductive and stretchable silver nanowire conductors." *Advanced materials* 24.37 (2012): 5117-5122.

Now the reasons why the reported H concentration is wrong:

- Thermodynamics. Sieverts law allows us to compute the equilibrium H concentration under the charging conditions used in this study (NB: such a calculation should have been performed and reported). The authors charge in 50/50 Ar/H₂ atmosphere. Although they do not say so explicitly, I take it the atmosphere was at room temperature and pressure. The formation energy of H interstitials in Ag is not reported by the authors. However, I know it to be larger than that of H in Ni, which is 0.17eV. Thus, I'll use the Ni value to come up with an upper bound estimate for the equilibrium H concentration. Following the expressions used by Bechtle et al. (*Acta Mater.* 57, 4148 [2009]), I arrive at an expected H concentration of 4 appm (~0.04 wppm). This value is orders of magnitude smaller than the ones reported by the authors (up to 75 wppm, i.e. ~8000 appm), which are high even when compared to those achievable under high pressure/high temperature H charging. For example, to achieve a H concentration of 3400 appm in Ni, Bechtle et al. had to charge in a pure H₂ atmosphere at 138 MPa and 200 C. In short, the H concentrations reported here are unrealistic for the charging conditions used.

Response: We agree that the hydrogen concentration is quite low in bulk silver. However, NWs can be fundamentally different from bulk samples. Due to the large surface/volume ratio of NWs, hydrogen in our system can stay near the NW surface.

If we only consider monolayer hydrogen adsorption on Ag surface, the area of each possible adsorption site s_H on different FCC surfaces is ka^2 , where a is the lattice constant and $k = 0.2 \sim 1$ is a coefficient depending on the surface orientation [1]. The volume per Ag atom in FCC lattice v_{Ag} equals $a^3/4$. If we approximately treat the Ag NW cross section as a circle with radius of r , we can estimate the adsorption amount of hydrogen on the NW surface by:

$$c_H = \frac{m_H}{m_{Ag}} \frac{N_H}{N_{Ag}} = \frac{m_H}{m_{Ag}} \frac{2\pi r L / s_H}{\pi r^2 L / v_{Ag}} = \frac{m_H}{m_{Ag}} \frac{a}{2kr}$$

where $m_H = 1.00794 \text{ u}$ and $m_{Ag} = 107.8682 \text{ u}$ are the atomic masses of hydrogen and silver atoms, respectively; N_H, N_{Ag} are the total numbers of hydrogen and silver atoms, respectively; L is the NW length; $a = 0.408 \text{ nm}$ is the lattice constant of silver; $s_H = ka^2$ is the surface area per adsorption site and $v_{Ag} = a^3 / 4$ is the volume per Ag atom.

For a NW with diameter of 70 nm, which leads to $r = 35 \text{ nm}$, we can estimate the hydrogen concentration to be around 54-272 wt ppm for high coverage on surface. This estimated hydrogen concentration is comparable to our experimental data.

[1] Daw, M. S. and Baskes, M. I., "Embedded-atom method: Derivation and application to impurities, surfaces, and other defects in metals." *Physical Review B* 29.12 (1984): 6443.

- Kinetics. In their response to one of the reviewers, the authors report a H diffusivity in Ag of $1e-14 \text{ m}^2/\text{s}$. The diameter of the Ag NWs is about 20nm. Thus, using d^2/D , we find that the characteristic diffusion time of H into and out of the NWs is less than 1s (0.04s, for what it's worth). Thus, for all practical experimental purposes, the H concentration in the NWs comes to equilibrium with the surrounding atmosphere instantaneously. By contrast, the variations in measured H concentration reported in supplementary figure 1 occur over times ranging up to 48 hours. Thus, these variations cannot be ascribed to the charging of the NWs. Furthermore, with a characteristic outgassing time of less than 1s, it is safe to say that any H that may have been charged into the NWs is completely gone by the time the mechanical tests are performed.

Response: Hydrogen adsorption in metal involves two steps: dissociation of H_2 molecule and transport of the chemisorbed hydrogen [1]. In our case, the dissociative adsorption of hydrogen on the Ag surface is the controlling step, not the diffusion of H atoms in NW.

Flat and clean Ag surface is known to have a very high energy barrier for hydrogen dissociation and formation of an Ag-H adsorptive bond is endothermic [1-3]. Literature shows that impurities atoms, such as surface and subsurface oxygen species, can promote dissociation of hydrogen molecules [4-5]. In the Ag NWs, the surface is not flat (e.g. roughness and surface steps/defects [6]). These conditions make hydrogen adsorption possible but the activation energy is still very high when compared with other FCC metals, like Cu or Pt [2-3]. The characteristic time scale of adsorption can be hours for this high activation energy of dissociation (see the estimated rate in Table.2 of [5]).

For desorption of hydrogen, we do not have direct experimental evidence of how much hydrogen was left on the NW surface when the mechanical tests were performed. However, sites of surface defects such as roughness and minor steps are known to have higher binding energy to hydrogen atoms [7-8]. For hydrogen atoms diffused to and trapped at these sites, desorption is less likely to occur when compared with a perfect metal surface. The suppression of surface dislocation nucleation as observed in experiments can be attributed to those hydrogen atoms that remain adsorbed on the NW surface especially trapped at the defect sites. Our atomistic simulations confirmed that the hydrogen atoms near the NW surface (Figure R2) and at the surface steps (Figure 4 in the manuscript) can suppress surface dislocation nucleation.

[1] Wandelt, K., ed. *Surface and Interface Science: Solid Gas Interfaces II*. Vol. 6. John Wiley & Sons, 2015.

- [2] Gomez, T., et al., "Reactivity of transition metals (Pd, Pt, Cu, Ag, Au) toward molecular hydrogen dissociation: extended surfaces versus particles supported on TiC (001) or small is not always better and large is not always bad." *The Journal of Physical Chemistry C* 115.23 (2011): 11666-11672.
- [3] Greeley, J. and Mavrikakis, M., "Surface and subsurface hydrogen: Adsorption properties on transition metals and near-surface alloys." *The Journal of Physical Chemistry B* 109.8 (2005): 3460-3471.
- [4] Mohammad, A. B., et al., "A computational study of H₂ dissociation on silver surfaces: The effect of oxygen in the added row structure of Ag (110)." *Physical Chemistry Chemical Physics* 9.10 (2007): 1247-1254.
- [5] Xu, Y., Greeley, J. and Mavrikakis, M., "Effect of subsurface oxygen on the reactivity of the Ag (111) surface." *Journal of the American Chemical Society* 127.37 (2005): 12823-12827.
- [6] Ramachandramoorthy, R., et al., "Reliability of single crystal silver nanowire-based systems: stress assisted instabilities." *ACS nano* 11.5 (2017): 4768-4776.
- [7] Myers, S. M., et al. "Hydrogen interactions with defects in crystalline solids." *Reviews of Modern Physics* 64.2 (1992): 559.
- [8] Christmann, K. and Ertl, G., "Interaction of hydrogen with Pt (111): the role of atomic steps." *Surface Science* 60.2 (1976): 365-384.

- Measurement method. The authors repeatedly state (in the manuscript and in their responses to the reviewers) that the best method of measuring H concentrations is SIMS. However, it is important to realize that no flavor for SIMS (including in situ TOF-SIMS) has an areal footprint smaller than about a square micron. Indeed, most have footprints that are much larger than that. By contrast, a vertically aligned, 20nm-diameter NW presents a cross-section area of three ten-thousandths of a square micron! The signal returned by SIMS from a measurement performed upon such a NW will be utterly dominated by the surrounding background (presumably, the substrate upon which the NW was grown). Most NWs are grown as forests, so perhaps a square micron-sized SIMS measurement will encompass several NWs, but that will still not lead to a meaningful NW signal in the SIMS measurement unless the spacing between the NW is comparable to (ideally: smaller than) the NW diameter. The authors do not report the spacing between NWs in their samples, but based on reports by others, I expect that it is much larger than the NW diameter. Thus, SIMS is not suitable for measuring H concentration in NWs. In the present study, it most likely measures H uptake into the substrate.

Response: Please see our response above on the Ag NW samples for the SIMS experiments. During the ion etching (SIMS), we can detect clear signals of H and Ag secondary ions from the NW network but not from the Si substrate. Ideally we would like to measure the hydrogen concentration directly on the same single NW used for mechanical testing. Unfortunately, up to now, there is no feasible approach to do so. SIMS is still a suitable method to detect the lightest element, hydrogen, in materials.

In light of the foregoing, I do not think that the narrative about hydrogen embrittlement presented in this manuscript is credible. On the other hand, the mechanical tests (which are impressive in their own right) clearly do show that something changes in the mechanical response of the NWs.

This is an interesting finding, but I think more work is needed to determine the reasons for these changes. Similarly, the modeling work is top notch. However, for the reasons presented above, I do not believe it is relevant to the mechanical tests.

Response: Based on our atomic simulations, hydrogen atoms from surface adsorption can suppress surface dislocation nucleation effectively and lead to similar hydrogen embrittlement in the charged NWs. See Figure R2 as an example. In addition, hydrogen atoms at the surface steps (Figure 4 in the manuscript) were found to effectively suppress surface dislocation nucleation.

Figure R2. Stress-strain curve with hydrogen only close to the NW surface.

This manuscript underwent thorough review by three other referees. All three raised the same concern about the H concentration as I did. However, unlike the previous reviewers, I do not find the authors' responses to these criticisms to be satisfactory.

However, for what it's worth, I think the authors' responses to all the other concerns are entirely adequate. In particular, I do not share the second reviewer's view that the work presented by the authors lacks novelty.

Response: We would like to express our sincere thanks to the reviewer's positive assessment of our work as well as his critical comments on the H concentration. We hope our above responses have satisfactorily addressed his/her concern about the measurement of hydrogen concentration in the charged samples and justify our overall interpretation of the hydrogen embrittlement in Ag NWs. In particular, we appreciate that the reviewer found that our responses to all other concerns are entirely adequate especially the second reviewer's view about the lack of novelty.

Reviewers' comments:

Reviewer #4 (Remarks to the Author):

The authors have provided a comprehensive response to my previous set of comments. They now state that hydrogen is not absorbed into Ag NWs in their experiment, but rather is adsorbed onto NW surfaces. To substantiate this interpretation, they estimate the maximum possible apparent H concentration based on full surface coverage of the NWs, demonstrating that it is of the order (or somewhat higher) than the maximum concentration they obtained from their SIMS measurements. Furthermore, they state that the slow uptake of H is due to a high energy barrier for H₂ dissociation on Ag surfaces.

The interpretation proposed by the authors is much more plausible than in the previous version of this manuscript. However, some issues nevertheless remain to be resolved, as listed below.

Concerning experimental data:

- the interpretation based on adsorption resolves my concern regarding the kinetics of H uptake. However, the thermodynamic question remains. To obtain near-complete surface coverage at the low H₂ pressures and temperatures used in this study, the adsorption energy must be quite low (even if the adsorption barrier is high). Is there evidence for this in previous work? Or can the authors provide evidence, e.g. using DSC experiments or DFT calculations?
- the interpretation of the experiments in terms of H surface adsorption is now key to the manuscript. However, it is reported in the supplementary materials, rather than in the manuscript body. I considered this arrangement to be inappropriate and to be revised.
- since the H is thought to reside on NW surfaces, it does not make sense for the authors to express H content in terms of wppm. Rather, they should present NW surface coverage fractions.

Concerning simulations: now that the experiments have been re-interpreted in terms of surface adsorption, the relevance of the atomistic modeling presented in the manuscript has been greatly reduced. All of that modeling has assumed that the H is distributed in the lattice, not on the surface. The authors have added a new simulation and figure in supplementary materials (SF. 7) to suggest that this discrepancy may be ignored. However, the model shown in SF. 7 does not correspond at all to H adsorbed on the surface, as the community usually conceives of it. On the contrary, all the H shown in that model resides in the subsurface regions. If the authors wish to claim that SF. 7 shows how H adsorption really works in Ag NWs, then they need to invest significant extra effort to make this argument. More likely, they need to repeat much of the modeling work using models that contain H adsorbed on NW surfaces, rather than trapped in NW interiors.

List of changes:

All the changes in manuscript have been highlighted in blue. The details of changes are:

1. In page 5, added “(equivalent to surface coverage θ_H from 0.17 to 0.67 for NW with diameter of 70nm)”
2. In page 5, added a paragraph “The high concentration of hydrogen ... ”
3. In page 6, added “ $\theta_H = 0.67$ ” at line 14.
4. In page 7, added “ $\theta_H = 0.67$ ” and “ $\theta_H = 0.33$ ” at line 3 and line 6.
5. In page 7, line 21 was revised as “Hydrogen atoms were initially inserted on the free surface of NW samples at random.”
6. In page 8, line 12 was revised as “The simulations indicate that hydrogen atoms initially adsorbed on the surface or charged in subsurface regions (Supplementary Fig. 9) can suppress ...”
7. In page 10, line 14 was revised as “Different concentrations of hydrogen were initially inserted on the free surface of the NW samples”
8. In page 19, line 6 was revised as “Hydrogen atoms were initially inserted on the free surface of NW samples at random.”
9. In page 19, added “5 different samples were tested for each hydrogen concentration.” at line 11.
10. In page 19, line 14 was revised as “Hydrogen atoms were initially inserted on the free surface of NW samples at random.”
11. Figure 2 was replaced based on new simulations.
12. Figure 3c was replaced based on new simulations.

In Supplementary Information:

1. In page 3, added “In our experimental conditions, the maximum hydrogen atoms absorbed on the NW surface (for a NW diameter of 70nm) is 8 atoms/nm² (Supplementary Table 1), about 2/3 (surface coverage, θ_H) of the (001) Ag surface (maximum around 12 atoms/nm² on (100) surface).”
2. In page 4, modified Supplementary Table 1, added converted surface adsorption data of hydrogen.
3. In page 5, added Supplementary Discussion “S2. DFT calculation of adsorption energy of Hydrogen on (001) Ag surface”, including Supplementary Figure 2-3 and Supplementary Table 2.

Response to the Reviewer Comments:

Reviewer #4 (Remarks to the Author):

The authors have provided a comprehensive response to my previous set of comments. They now state that hydrogen is not absorbed into Ag NWs in their experiment, but rather is adsorbed onto NW surfaces. To substantiate this interpretation, they estimate the maximum possible apparent H concentration based on full surface coverage of the NWs, demonstrating that it is of the order (or somewhat higher) than the maximum

concentration they obtained from their SIMS measurements. Furthermore, they state that the slow uptake of H is due to a high energy barrier for H₂ dissociation on Ag surfaces. The interpretation proposed by the authors is much more plausible than in the previous version of this manuscript. However, some issues nevertheless remain to be resolved, as listed below.

Concerning experimental data:

- the interpretation based on adsorption resolves my concern regarding the kinetics of H uptake. However, the thermodynamic question remains. To obtain near-complete surface coverage at the low H₂ pressures and temperatures used in this study, the adsorption energy must be quite low (even if the adsorption barrier is high). Is there evidence for this in previous work? Or can the authors provide evidence, e.g. using DSC experiments on DFT calculations?

Our response:

To answer this question, the adsorption energies of H atoms on (001) Ag surface and close to a surface ledge are calculated using DFT. The detail of the calculation and results are described in the following part and also included in the SI:

To compute the adsorption energy of hydrogen on Ag surface, we use the DFT method implemented in the Vienna ab initio simulation package (VASP)¹. The pseudopotentials used are of the PAW² type and the exchange correlation energy is evaluated using the Perdew-Burke-Ernzerhof³ generalized gradient approximation (GGA). The electronic wave functions were represented in a plane-wave basis set with energy cut-off of 400 eV. 6x6x1 Monkhorst Pack integration scheme in k space were used. Residual forces after relaxation are smaller than 0.01 eV/Å. Dipole-dipole interactions are corrected for the calculation of energies.

The (001) Ag surface was considered for hydrogen adsorption. A slab of Ag was created with 8 atomic layers in the thickness direction. A 16 Å thick vacuum layer was put along the z-direction to mimic the free surface in the simulation cell. The cell volume and the two very bottom atomic layers were fixed during the calculations.

Adsorption energies were computed by subtracting the energies of the adsorbate atoms and the slab from the energy of the adsorbates/slab system as shown in the following equation:

$$n\Delta E_{ad} = E_{Ag(001)H_n,ads} - E_{Ag(001)} - E_{H_n}$$

where $E_{Ag(001)H_n,ads}$ is the total energy of the system with adsorbed H_n , $E_{Ag(001)}$ is the energy of the Ag (001) slab, and E_{H_n} is the energy of the adsorbates. With this definition, a negative ΔE_{ad} indicates stable adsorption on the slab.

The adsorption energies of an H atom were calculated for the high-symmetry sites on the Ag surfaces, including top, bridge and four-fold hollow positions shown in Fig. R1. The adsorption energies on the high-symmetry sites close to a surface ledge are also considered, as shown in Fig. R2.

Fig. R1 | High-symmetry sites for adsorption of an H atom on Ag (001) surface (top view). (a) 4-fold Hollow site. (b) Bridge site. (c) Top site. Pink and white spheres represent first and second layer atoms, respectively.

Fig. R2 | High-symmetry sites for adsorption of an H atom on Ag (001) surface close to a surface ledge. (a) Side view of the atomistic model. The atoms are colored by the relative position in the height direction. (b-d) Top view of the model. (b) Bridge site. (c) Top site. (d) 4-fold Hollow site.

Table R1. Adsorption energy of hydrogen adsorbate on (001) Ag surface.

	Adsorption site	Adsorption Energy (eV)
On (001) Ag surface	Top	-2.097
	Bridge	-3.030
	4-fold Hollow	-3.023

Below (001) Ag surface	Octahedral interstitial site	-2.549
Above (001) Ag surface with ledge	Top	-3.165
	Bridge	-3.032
	4-fold Hollow	-3.017

The calculated adsorption energies of hydrogen adsorbate on (001) Ag surface are shown in Table R1. Although the adsorption barrier is high when considering the bond energy of the H₂ molecule, the calculated adsorption energies at all the high-symmetry sites are negative, indicating stable adsorption on the metal surface. In addition, when a ledge exists on the free surface, as shown in Fig. R2a, the lowest adsorption energy further drops to -3.165 eV on the top site as shown in Table R1. The interstitial energy of hydrogen atom was also calculated and included in Table R1. The system has lower energy when hydrogen is adsorbed on bridge or 4-fold hollow surface sites than in an octahedral interstitial site. Previous studies indicate that impurity atoms, such as surface and subsurface oxygen species, can further decrease the adsorption energy^{4, 5}. The DFT calculations confirm stable adsorption of hydrogen adsorbate on (001) Ag surface.

- the interpretation of the experiments in terms of H surface adsorption is now key to the manuscript. However, it is reported in the supplementary materials, rather than in the manuscript body. I considered this arrangement to be inappropriate and to be revised.

Our response:

We have revised the manuscript and added the interpretation of the experiments in terms of H surface adsorption in the main text.

- since the H is thought to reside on NW surfaces, it does not make sense for the authors to express H content in terms of wppm. Rather, they should present NW surface coverage fractions.

Our response:

For comparison, we listed the surface adsorbed H atoms in terms of H atoms/nm² and surface coverage θ_H for a typical NW diameter of 70 nm in Supplementary Table 1. In our experimental conditions, the maximum hydrogen atoms adsorbed on the NW surface (for a NW diameter of 70nm) is 8 atoms/nm² (Supplementary Table 1), covered about 2/3 of the (001) Ag surface (the maximum is around 12 atoms/nm² in (100) surface).

In the main text, we decided to keep both the H concentrations c_H in the unit of wt ppm as well as the surface coverage θ_H since wt ppm is directly measured from the experiment and the calculated surface absorbed values are depending on the averaged NW diameter.

Concerning simulations: now that the experiments have been re-interpreted in terms of surface adsorption, the relevance of the atomistic modeling presented in the manuscript has been greatly reduced. All of that modeling has assumed that the H is distributed in the lattice, not on the surface. The authors have added a new simulation and figure in

supplementary materials (SF. 7) to suggest that this discrepancy may be ignored. However, the model shown in SF. 7 does not correspond at all to H adsorbed on the surface, as the community usually conceives of it. On the contrary, all the H showed in that model resides in the subsurface regions. If the authors wish to claim that SF. 7 shows how H adsorption really works in Ag NWs, then they need to invest significant extra effort to make this argument. More likely, they need to repeat much of the modeling work using models that contain H adsorbed on NW surfaces, rather than trapped in NW interiors.

Our response:

To eliminate the discrepancy between atomic simulations and experiments, according to the new surface adsorption model, we have performed additional simulations with all the initial hydrogen atoms adsorbed on NW free surfaces, as shown in the Fig. R3.

Fig. R3 | Simulations with H initially only on NW free surface. Red color stands for Ag atoms and blue color stands for hydrogen atoms.

We have replaced Figure 2 in the manuscript with the new simulation results. Five different samples containing 57 wt ppm hydrogen only adsorbed on surface were stretched at a constant strain rate of $10^6/s$ under NVT ensemble. The localized dislocation nucleation and necking can still be captured in the simulations as shown in Fig. R4a, and yield strength is higher than that of the hydrogen-free NW, consistent with our experimental and previous simulation results.

Fig. R4 | MD simulation of tensile testing of penta-twinned Ag NWs. a, Simulated transition from distributed plasticity in the absence of hydrogen to localized plasticity in the presence of hydrogen. Scale bar, 7 nm. b, Averaged stress-strain curves of Ag NWs at different hydrogen concentrations with error bars.

We have also re-done the stress-relaxation simulations in Figure 3c with all the initial hydrogen on the free surface of NW samples. The new results still clearly show that the presence of hydrogen will delay and suppress dislocation nucleation in penta-twinned Ag NWs, which is consistent with previous simulations. The origin Figure 3 has been replaced by the new Figure as shown in Fig. R5 here.

Fig. R5 | *In situ* TEM tests and MD simulations of stress relaxation. **a**, Stress-strain curves from stress relaxation tests for penta-twinned Ag NWs at different concentrations of hydrogen. **b**, Experimentally measured stress relaxation and associated strain evolution curves for penta-twinned Ag NWs at different hydrogen concentrations. **c**, MD simulations of stress relaxation in penta-twinned Ag NWs at different hydrogen concentrations. Note that the all the hydrogen atoms are initially on the NW surface. Red atoms in the grain show stacking faults in the right insets.

We have revised the method part too.

Reviewers' Comments:

Reviewer #4:

Remarks to the Author:

The authors have addressed all remaining concerns and their article is now suitable for publication.